# Value-Guided KV Compression for LLMs via Approximated CUR Decomposition

**Ayan Sengupta**[*]**, Siddhant Chaudhary**[*] **& Tanmoy Chakraborty**
Department of Electrical Engineering
Indian Institute of Technology Delhi, India
ayan.sengupta@ee.iitd.ac.in, urssidd@gmail.com, tanchak@iitd.ac.in

## Abstract

Key-value (KV) cache compression has emerged as a critical technique for reducing the memory and latency overhead of autoregressive language models during inference. Prior approaches predominantly rely on query-key attention scores to rank and evict cached tokens, assuming that attention intensity correlates with semantic importance. However, this heuristic overlooks the contribution of value vectors, which directly influence the attention output. In this paper, we propose CurDKV, a novel, value-centric KV compression method that selects keys and values based on leverage scores computed from CUR matrix decomposition. Our approach approximates the dominant subspace of the attention output $\mathrm{softmax}(QK^\top)V$, ensuring that the retained tokens best preserve the model's predictive behavior. Theoretically, we show that attention score approximation does not guarantee output preservation, and demonstrate that CUR-based selection minimizes end-to-end attention reconstruction loss. Empirically, CurDKV achieves up to $9.6\%$ higher accuracy than state-of-the-art methods like SnapKV and ChunkKV under aggressive compression budgets on LLaMA and Mistral, while maintaining compatibility with FlashAttention and Grouped Query Attention. In addition to improved accuracy, CurDKV reduces generation latency by up to 40% at high compression, offering a practical speed-accuracy tradeoff.

## 1 Introduction

Transformer-based large language models (LLMs) have achieved remarkable performance across a wide range of natural language understanding and generation tasks [Yang et al., 2024, Grattafiori et al., 2024, Jiang et al., 2023]. However, their inference-time memory consumption remains a significant challenge, particularly for application requiring long-context information. A major source of this overhead is the *Key-Value (KV) cache*, which stores the key and value vectors corresponding to previously generated tokens. These vectors are used to compute attention outputs without recomputing intermediate states, but their memory footprint grows linearly with sequence length. For instance, as highlighted by Feng et al. [2025], LLM-8B, which generats a sequence of 2M tokens, can consume up to 256GB of GPU memory while storing the KV cache.

Typically, KV cache is populated in two stages: during the *prefill phase*, where a long-context input is encoded in parallel and all token-level KV vectors are stored; and during the *generation phase*, where tokens are produced autoregressively and KV entries are appended one step at a time. In particular, the prefill phase dominates memory usage, especially when handling large input contexts.

---

[*]These authors contributed equally to this work.

39th Conference on Neural Information Processing Systems (NeurIPS 2025).

This has motivated extensive research [Li et al., 2024a] into KV compression techniques that reduce the number of cached entries while preserving output quality. Existing KV compression methods prioritize tokens based on attention scores. For instance, SnapKV [Li et al., 2024b] and H2O [Zhang et al., 2023] estimate token importance by accumulating attention weights across heads and layers. While such heuristics are computationally cheap, they only capture query-key alignment and neglect the downstream contribution of the value vectors. Moreover, these methods mostly overlook the fact that tokens with low attention scores can still carry rich semantic information through their value vectors. Recent work formalizes this mismatch through the concept of *eviction loss* [Feng et al., 2025], which quantifies the performance degradation from evicting specific KV entries. Critically, eviction loss is not always correlated with attention score (c.f. Figure 1), highlighting the need for value-aware selection strategies.

In this work, we introduce CurDKV (**CUR D**ecomposition for **KV** Compression), a principled and efficient KV compression method that circumvents these limitations by leveraging the structure of the *original value and key matrices* rather than relying on attention scores. CurDKV assigns importance to each key and value based on its *leverage score*, computed as the squared $\ell_2$ norm of the corresponding left-singular vector of the matrix in consideration. These scores reflect each token's contribution to the attention output and naturally align with the goal of minimizing eviction loss. To improve scalability, we also introduce a fast approximation based on *random Gaussian projections*, where the value and key matrices are projected into a lower-dimensional subspace before computing leverage scores. This maintains relative importance across tokens while significantly reducing computational cost. We further cast the KV se-

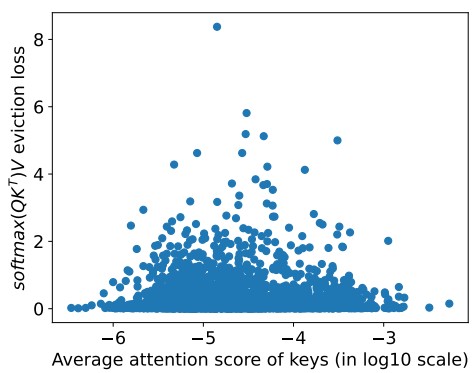

Figure 1: Keys associated with high average attention value do not necessarily have high post-eviction $softmax(QK^\top)V$ reconstruction (eviction) loss.

lection problem as a *CUR decomposition* task on the attention matrix product, selecting key and value vectors that best reconstruct either the attention matrix $QK^\top$ or the final output $\mathrm{softmax}(QK^\top)V$. Unlike prior work, our formulation directly targets the fidelity of the attention output, leading to a more semantically consistent compression strategy. As shown in Figure 2, CurDKV achieves significantly lower reconstruction loss on both the $QK^\top$ matrix and the final $\mathrm{softmax}(QK^\top)V$ product, demonstrating improved preservation of contextual semantics post-compression.

We evaluate CurDKV on two popular long-context benchmarks – LongBench [Bai et al., 2023] and Ruler [Hsieh et al., 2024], spanning 24 tasks with LLaMA-3.1-8B-Instruct [Grattafiori et al., 2024] and Mistral-7B-Instruct [Jiang et al., 2023]. Our experiments demonstrate that CurDKV consistently outperforms existing attention-based KV compression baselines such as SnapKV [Li et al., 2024b], ChunkKV [Liu et al., 2025], and Streaming LLM [Xiao et al., 2024]. In particular, CurDKV achieves significantly lower degradation in generation accuracy at aggressive compression ratios (e.g., $90\%$ cache reduction). With LLaMA, CurDKV surpasses SnapKV by up to $9.6\%$ in average task score, and achieves higher fidelity (similarity as the full cache model) across all types of long-context-dependent tasks. Similar gains are observed with Mistral, where CurDKV maintains over $95\%$ average fidelity even under constrained memory budgets. These results validate that our value-centric, CUR-based strategy more effectively preserves semantic attention outputs than methods relying solely on query-key attention scores.[1]

## 2 Related Work

**KV Cache compression via attention heuristics and top-$k$ eviction.** A widely used class of KV compression methods rank cached tokens by heuristics and retain the top-$k$ scoring entries. H2O [Zhang et al., 2023] uses observed attention scores from specific query-key interactions to

---

[1]We have uploaded the source code and datasets as supplementary; we are committed to release them upon acceptance of the paper.

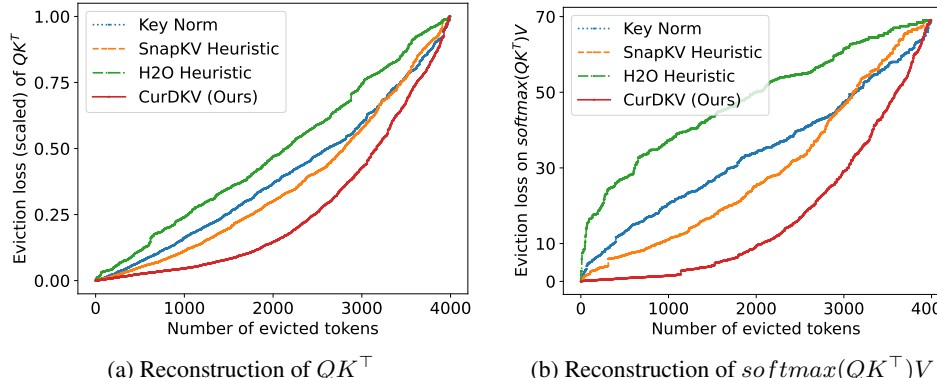

(a) Reconstruction of $QK^\top$

(b) Reconstruction of $softmax(QK^\top)V$

Figure 2: Preservation of the $QK^\top$ and $\mathrm{softmax}(QK^\top)V$ matrices under different KV compression strategies. `CurDKV` achieves the lowest eviction loss, indicating more faithful reconstruction of intermediate attention. We compute the reconstruction loss for LLaMA-3.1-8B-Instruct, averaged over all layers and attention heads for a sample text obtained from `https://en.wikipedia.org/wiki/Attention_Is_All_You_Need`.

estimate per-token importance and prunes less relevant tokens online. However, it requires access to attention weights, which breaks compatibility with efficient kernels like FlashAttention [Dao et al., 2022]. SnapKV [Li et al., 2024b] computes cumulative attention scores and selects the highest-ranked tokens, assuming that frequently attended tokens are most informative. Scissorhands [Liu et al., 2023c] further introduces temporal windows to detect consistently attended tokens. PyramidKV [Cai et al., 2024] proposes a hierarchical scoring scheme across layers and heads, combining multi-level top-$k$ selection with scaling-aware normalization. ChunkKV [Liu et al., 2025] compresses the cache by segmenting it into chunks and retaining only top-ranked entries per chunk to ensure temporal coverage. While these approaches are efficient and easy to deploy, they focus on approximating $QK^\top$ and neglect the downstream contribution of value vectors in $\mathrm{softmax}(QK^\top)V$, the final attention output propagated to the subsequent layers. Beyond static heuristics, adaptive methods attempt to allocate compression budgets more intelligently across heads or layers. AdaKV [Feng et al., 2025] computes attention statistics such as entropy and average weights to estimate the relative importance of each head and distributes the token retention budget accordingly. While more flexible, these approaches still rely on attention-derived metrics and do not explicitly account for the downstream impact of compression on the attention output.

**Compatibility with efficient attention implementations.** A critical practical consideration is compatibility with modern inference backends such as FlashAttention [Dao et al., 2022], which avoids materializing the full attention matrix by fusing softmax and matrix multiplications. Methods such as H2O [Zhang et al., 2023], which rely on observed attention weights, are incompatible with FlashAttention and similar kernels that trade explicit attention computation for efficiency. Another major limitation with the majority of the existing KV compression methods is their inability to work with Group Query Attention (GQA). GQA [Ainslie et al., 2023] has been widely adopted in recent LLM architectures such as LLaMA [Grattafiori et al., 2024] and Mistral [Jiang et al., 2023] due to its ability to reduce the memory footprint of the KV cache by sharing keys and values across multiple heads. However, most existing KV cache compression techniques, including SnapKV and PyramidKV, do not support GQA natively. These methods redundantly replicate the compressed KV cache across heads within a group, thus forfeiting GQA's inherent efficiency. To address this limitation, Feng et al. [2025] proposed a GQA-compatible eviction framework that computes average attention weights within each group to identify important tokens. This enhancement enables state-of-the-art KV compression methods to function effectively in GQA-enabled models and achieve significant cache size reductions while preserving generation quality.

# 3 Methodology

## 3.1 Preliminaries

**KV Cache.** The generation process of an LLM is autoregressive in nature, i.e., each generation step relies on the outputs of all previous steps. Suppose $X \in \mathbb{R}^{n \times d}$ be the matrix containing the hidden states of a layer in an LLM upto a time step. Further, suppose $x \in \mathbb{R}^{1 \times d}$ be the hidden state of the last generated token, which is used as an input for the current generation step. Suppose, there are $h$ attention heads. For a head $i \in [1, h]$, the query, key and value matrices, $W_i^Q$, $W_i^K$ and $W_i^V$ (all in $\mathbb{R}^{d \times d_h}$) map hidden states $X$ to query, key and value states; specifically, the following are computed:

$$Q_i = XW_i^Q, K_i = XW_i^K, V_i = XW_i^V \tag{1}$$

For the current time step, the states $K_i$ and $V_i$ constitute the KV cache elements for head $i$. *KV caching* refers to caching $K_i$ and $V_i$ in memory to avoid recomputing them at every generation step. After computing the output for the current time step, the KV cache is updated accordingly:

$$K_i = Cat[K_i : xW_i^K], V_i = Cat[V_i : xW_i^V] \tag{2}$$

KV cache compression techniques aim to retain only the most relevant key and value states from $K_i$ and $V_i$, without significant loss in generation quality. Particularly, we show that the loss on the attention output is upper bounded by the Frobenius distance between the original value matrix and it's compression:

**Lemma 3.1.** *Let $Q, K, V \in \mathbb{R}^{n \times d}$ be the query, key and value matrices in the attention computation. Further, let $K', V' \in \mathbb{R}^{n \times d}$ be sub-matrices obtained by zeroing-out a subset of rows of $K$ and $V$ respectively. Then,* [2]

$$\|softmax(QK^T)V - softmax(QK'^T)V'\|_F \leq \sqrt{n}\|V - V'\|_F + 2\sqrt{n}\|V'\|_F \tag{3}$$

Note that for higher compression levels, the second term ($\|V'\|_F$) becomes negligible; in that case, the role of the approximation $V'$ to $V$ becomes significant. On the other hand, the contribution of $Q$, $K$ and $K'$ is absorbed by the softmax operator. This motivates us to design a *value-guided* compression technique.

**Low-rank decompositions.** Let $A \in \mathbb{R}^{n \times d}$ be a matrix. The problem of *low-rank decomposition* of $A$ seeks to factor $A$ as a product of low-rank matrices, and is a well-studied problem with wide applications in mathematical modeling and data compression [Halko et al., 2010]. The optimal solution to this problem is the well-known *singular value decomposition* (SVD), which represents $A$ in the form $A = U\Sigma V^*$, with $U$ and $V$ being semi-unitary matrices, and $\Sigma$ a square diagonal matrix of dimension equal to the rank of $A$. While the SVD is widely adopted in data analysis problems, when applied to the context of the KV cache compression problem, it presents an inherent challenge; namely, the column/row vectors of $U$ and $V$ are not representative of the column/row vectors of $A$ itself. Another suboptimal low-rank decomposition, called the *CUR decomposition*, resolves this limitation: it is an approximation of the form $A \approx CUR$, where $C$ is a subset of columns of $A$, $R$ is a subset of rows of $A$ and $U$ is a low-rank matrix. Here, while the matrices $C$ and $R$ are representative of the underlying data, the matrix $U$ optimizes the approximation to $A$. The CUR decomposition is known to be a good low-rank approximation to the original matrix [Woodruff, 2014, Halko et al., 2010, Mahoney and Drineas, 2009].

**Computation of CUR and Leverage Scores.** We utilize *leverage scores*, a popular technique used to compute CUR factorizations based on identifying key elements of the singular vectors of a matrix [Mahoney and Drineas, 2009]. As before, let $A = U\Sigma V^*$ be the SVD of $A \in \mathbb{R}^{n \times d}$, where $U \in \mathbb{R}^{n \times r}$, $\Sigma \in \mathbb{R}^{r \times r}$ and $V \in \mathbb{R}^{d \times r}$ with $r = \text{rank}(A)$. The idea of *leverage scores* is to compute importance scores for each row and column of $A$, which are then used to sample the rows and columns that then constitute the matrices $R$ and $C$ of the CUR decomposition respectively. Particularly, to rank the importance of rows of $A$, we define the *row leverage scores* $l_{r,j}$ (for the $j$th row of $A$) as

$$l_{r,j} := \|U(j, :)\|_2 \tag{4}$$

---

[2]See section A of the Appendix for a proof.

Similarly, to rank the importance of columns of $A$, the *column leverage scores* $l_{c,j}$ (for the $j$th column of $A$) are defined analogously by

$$l_{c,j} := \|V^*(:,j)\|_2 = \|V(j,:)\|_2 \tag{5}$$

Since $U$ and $V$ contain the left and right singular vectors of $A$, it easily follows that $\sum_{j=1}^{n} l_{r,j}^2 = n$ and $\sum_{j=1}^{d} l_{c,j}^2 = d$. In turn, these scores lead to the distributions $\{\frac{l_{r,j}^2}{n}\}$ and $\{\frac{l_{c,j}^2}{d}\}$ from which the matrices $R$ and $C$ can be sampled. Alternatively, static methods use the top-$k$ scores from either distribution to sample the corresponding matrices.

## 3.2 `CurDKV`: Algorithm Design

We now introduce `CurDKV`, our proposed KV cache compression method, which uses combined leverage scores of key and value matrices to compress the KV matrix. `CurDKV` can be applied during the pre-filling as well as the token generation phase, is easy to implement, and seamlessly integrates with inference acceleration methods such as FlashAttention [Dao et al., 2022] and GQA [Ainslie et al., 2023]. Algorithm 1 provides the pseudocode of `CurDKV`. In the following discussion, we assume the attention setup as in GQA [Ainslie et al., 2023], wherein a group of attention heads uses a common key and value matrix. Particularly, we assume that there are $g$ groups, with the key and value matrices of group $i$, denoted by $K_i \in \mathbb{R}^{n \times d}$ and $V_i \in \mathbb{R}^{n \times d}$, respectively. As usual, $n$ represents the number of keys (or values) in consideration, and $d$ is the dimension of the space of vectors. The combined key and value matrices across all groups is denoted by $K \in \mathbb{R}^{g \times n \times d}$ and $V \in \mathbb{R}^{g \times n \times d}$.

As shown in Algorithm 1, `CurDKV` takes as input the key matrix $K \in \mathbb{R}^{g \times n \times d}$ across all attention groups, the corresponding value matrix $V \in \mathbb{R}^{g \times n \times d}$, the *group budget* $k \in [n]$ representing the level of compression, a projection dimension $r$ and the number of initial attention sinks $s$ to be preserved. It outputs compressed key and value matrices $K' \in \mathbb{R}^{g \times k \times d}$ and $V' \in \mathbb{R}^{g \times k \times d}$, which are then used to compute the output of the attention module. At a high-level, `CurDKV` computes *approximate leverage scores* for each *row* of the key and value matrices $K_i \in \mathbb{R}^{n \times d}$ and $V_i \in \mathbb{R}^{n \times d}$ within a single group $i$ (lines 3-5). The leverage scores for corresponding keys and values are then combined, followed by normalization (lines 6-7). Finally, the computed leverage scores are then used to retrieve the top-$k$ keys and values from $K_i$ and $V_i$ (lines 9-11). In addition, `CurDKV` also preserves the initial $s$ tokens (lines 8, 10) due to the prevalence of attention sinks [Xiao et al., 2024].

**Approximate leverage scores and Gaussian projections.** The computation of exact leverage scores as in Equations 4 and 5 requires the computation of the SVD of the matrices in consideration, which turns out to be a bottleneck for the KV compression problem. To that end, `CurDKV` projects $K_i$ and $V_i$ to matrices $K_i G$ and $V_i G$, respectively, where $G \in \mathbb{R}^{d \times r}$ is a random matrix, each of whose entries are sampled from a normal $\mathcal{N}(0, \frac{1}{r})$ distribution; in our implementation, we use $r = 20$ [3]. After this projection step, the norms of the rows of $K_i G$ and $V_i G$ are used as proxies for the actual leverage scores (lines 3-5 of Algorithm 1).

**Adaptive budget allocation for `CurDKV`.** Along with `CurDKV`, we also implement `AdaCurDKV`, an adaptive variation of `CurDKV`. `AdaCurDKV` adaptively achieves head-specific compression by choosing the top-$k$ key and value vectors *across all heads in a layer*, where the selection is based on the computed leverage scores. As in the standard adaptive implementation introduced by Feng et al. [2025], a safeguard parameter $\alpha$ is used to ensure a minimum fraction of key/value vectors are preserved for each head (a default value of $\alpha = 0.20$ is used in our implementation).[4]

# 4 Experiments and Results

## 4.1 Experimental Settings

For the empirical analyses, following the contemporary studies, we use two instruction-tuned LLMs – LLaMA-3.1-8B-Instruct [Grattafiori et al., 2024] and Mistral-7B-Instruct-v0.3 [Jiang et al., 2023].

---

[3]The Gaussian projection is motivated by the construction in Theorem 19 of Woodruff [2014].

[4]Refer to `https://github.com/NVIDIA/kvpress/blob/main/kvpress/presses/adakv_press.py` for an implementation of the adaptive compression logic.

**Algorithm 1:** `CurDKV`: CUR-based KV Compression with GQA Support

---

**Input:** Key matrix $K \in \mathbb{R}^{g \times n \times d}$, Value matrix $V \in \mathbb{R}^{g \times n \times d}$, group budget $k$, Gaussian projection dim $r$, num_sink $s$

**Output:** Compressed keys $K' \in \mathbb{R}^{g \times k \times d}$, values $V' \in \mathbb{R}^{g \times k \times d}$

---

**1 for** *each group $g_i$ in $1, \ldots, g$* **do**

**2** $\quad$ Sample Gaussian matrix $G \in \mathbb{R}^{d \times r}$ with $G_{ij} \sim \mathcal{N}(0, 1/r)$ ;

**3** $\quad$ Project $K_i \leftarrow K_i G$, $V_i \leftarrow V_i G$ ;

**4** $\quad$ Compute key leverage scores: $\ell_j^{(K)} = \|K_i[j]\|_2^2$, value leverage: $\ell_j^{(V)} = \|V_i[j]\|_2^2$ ;

**5** $\quad$ Combine scores: $\ell_j^{(KV)} = \ell_j^{(K)} \cdot \ell_j^{(V)}$ ;

**6** $\quad$ Normalize: $\tilde{\ell}_j = \ell_j^{(KV)} / \sum_j \ell_j^{(KV)}$ ;

**7** $\quad$ Preserve first $s$ tokens as sink indices: $S_{\text{sink}} = \{0, \ldots, s-1\}$ ;

**8** $\quad$ Select top-$(k-s)$ indices from $\tilde{\ell}[s:]$: $S_{\text{top}} = \text{TopK}(\tilde{\ell}[s:], k-s) + s$ ;

**9** $\quad$ Combine: $S_i = S_{\text{sink}} \cup S_{\text{top}}$ ;

**10** $\quad$ $K_i' \leftarrow K_i[S_i]$, $V_i' \leftarrow V_i[S_i]$ ;

**11 return** $K', V'$

---

All the evaluations are done on two widely popular long-context benchmarks – LongBench [Bai et al., 2023] and Ruler [Hsieh et al., 2024]. LongBench benchmark contains a total 16 tasks, covering various task domains including single-document QA, multiple-document QA, summarization, few-shot learning, synthetic tasks and code completion. From Ruler benchmark, we consider a total eight needle-in-a-haystack tasks with a maximum context length of 16K. Details of the LongBench and Ruler tasks and the prompt templates can be found in the appendix. For all these 24 tasks, we evaluate only on the more challenging question-agnostic setting [NVIDIA, 2024], where the questions are omitted during compression and only the context is compressed. As argued by Feng et al. [2025], this setup mimics more challenging scenarios, where the compression method is unaware of the questions being passed to the model during inference.

We compare `CurDKV` with various competitive KV compression baselines, including SnapKV [Li et al., 2024b], ChunkKV [Liu et al., 2025], LLM Streamline [Xiao et al., 2024] and Ada-SnapKV [Feng et al., 2025]. We consider an additional baseline, KNorm [NVIDIA, 2024] that uses key norm as a proxy to determine which keys to evict during compression. H2O [Zhang et al., 2023] is purposefully omitted from the evaluation as it throws out-of-memory error when run on a single NVIDIA A100-80GB GPU card (as it does not naively support flashattention, therefore requiring $4\times$ more cache memory on long sequences). All these baselines were run for different compression (eviction) ratios $\{30\%, 50\%, 70\%, 90\%\}$. Fol-

Table 1: Results of LLaMA-8B and Mistral-7B with full KV cache (0% compression) on LongBench.

| Task Type | Task | LLaMA-8B | Mistral-7B |
|---|---|---|---|
| Single-Doc QA | NrtvQA | 30.7 | 28.4 |
| | Qasper | 47.2 | 40.3 |
| | MF-en | 55.6 | 51.9 |
| Multi-Doc QA | HotpotQA | 59.5 | 48.9 |
| | 2WikiMQA | 51.8 | 37.4 |
| | Musique | 32.6 | 28.0 |
| Summarization | Gov Report | 35.0 | 34.7 |
| | QMSum | 25.1 | 25.5 |
| | Multi News | 26.8 | 26.8 |
| Few-shot Learning | TREC | 29.5 | 55.8 |
| | TriviaQA | 85.7 | 85.0 |
| | SAMSum | 38.7 | 20.8 |
| Synthetic | Pcount | 10.7 | 5.1 |
| | Pre | 100.0 | 98.0 |
| Code | Lcc | 53.9 | 49.8 |
| | RB-P | 47.6 | 56.1 |
| **Average** | | 45.7 | 43.3 |

lowing Xiao et al. [2024], we use attention sinks of size $s = 4$ for all the baselines (all the baseline numbers are produced in-house).

### 4.2 Results on LongBench

We report the results obtained with full cache (0% compression) for LLaMA-8B and Mistral-7B in Table 1. Table 2 summarizes the results obtained with LLaMA-8B and Mistral-7B with KV cache compressed at 30% and 90% with different baselines.

Table 2: KV compression methods on LongBench with LLaMA-3.1-8B-Instruct and Mistral-7B-Instruct (⬇: compression ratio). Compression ratio is the complement of the cache budget, *e.g.,* for a cache budget of 30% the corresponding compression ratio is 70%. Friedman statistics of 45.2/35.7 (for 30% compression) and 21.1/8.5 (90% compression) with p-values 0 indicate the statistical significance of the result obtained by CurDKV over the baselines. **Bold** highlights the best adaptive and non-adaptive baselines for each model and task, with blue highlighting the cases where CurDKV or AdaCurDKV are the best baselines. Results with other compression ratios are reported in Tables 14 and 15 in Appendix C. Each result cell "$x/y$" indicate accuracies with LLaMA/Mistral.

| ⬇ | Task | ChunkKV | Knorm | Streaming LLM | SnapKV | CurDKV | AdaSnapKV | AdaCurDKV |
|---|---|---|---|---|---|---|---|---|
| | | | | **Non-Adaptive Methods** | | | **Adaptive Methods** | |
| | NrtvQA | 30.3 / 25.7 | 30.6 / 23.8 | 26.8 / 24.4 | 30.2 / 26.4 | **31.7 / 28.8** | 29.7 / 25.4 | **31.9 / 27.6** |
| | Qasper | 45.7 / 36.3 | 44.4 / 35.3 | 43.4 / 36.1 | 46.5 / 35.5 | **48.2 / 39.4** | 45.8 / 35.7 | **48.6 / 41.4** |
| | MF-en | 50.2 / 48.7 | 55.2 / 48.7 | 36.0 / 34.4 | 53.8 / 49.0 | **56.3 / 52.9** | 55.3 / 48.4 | **56.9 / 51.0** |
| | HotpotQA | 58.0 / 49.0 | 57.2 / 48.7 | 52.1 / 43.9 | **58.4** / 48.8 | 57.8 / **49.2** | **59.9 / 47.3** | 59.7 / 46.6 |
| | 2WikiMQA | 49.6 / 37.3 | 49.4 / 36.6 | 42.5 / 31.6 | 49.3 / 37.2 | **51.3 / 41.5** | 50.1 / 37.2 | **52.5 / 39.6** |
| | Musique | 30.0 / 25.4 | 33.6 / 24.5 | 28.9 / 20.7 | 31.0 / 25.4 | **33.7 / 29.4** | 32.5 / 28.1 | **32.9 / 28.9** |
| | Gov Report | 33.9 / 34.1 | 34.0 / 34.0 | 31.7 / 32.9 | 33.6 / 33.6 | **34.5 / 35.9** | 34.1 / 34.1 | **35.0 / 34.7** |
| | QMSum | 24.0 / 24.6 | 24.7 / 24.2 | 23.5 / 23.8 | 23.9 / 24.2 | **25.0 / 25.1** | 24.6 / 24.8 | **25.2 / 25.3** |
| 30% | Multi News | 26.8 / 26.2 | 26.5 / 26.3 | 26.3 / 26.3 | 26.6 / 26.0 | **27.2 / 26.9** | 26.7 / 26.2 | **27.1 / 27.3** |
| | TREC | 25.0 / 50.9 | 66.0 / 49.3 | 32.5 / 59.5 | 31.0 / 51.5 | **67.0 / 60.0** | 30.5 / 52.0 | 63.5 / 73.0 |
| | TriviaQA | 85.5 / 84.8 | 87.8 / 85.6 | 91.7 / 76.2 | 86.4 / 85.4 | **92.2 / 89.0** | 86.4 / 88.1 | **92.6 / 89.3** |
| | SAMSum | 39.6 / 20.8 | 36.5 / 33.9 | 38.7 / 19.1 | 40.1 / 21.0 | **41.3 / 41.1** | 40.0 / 22.4 | **41.5 / 46.6** |
| | Pcount | 11.1 / 5.6 | 13.1 / 4.1 | 8.3 / 2.2 | 12.2 / 5.2 | **13.7 / 5.7** | **12.2 / 5.1** | 12.1 / 4.5 |
| | Pre | 99.5 / **97.5** | 98.0 / 80.8 | 68.5 / 68.0 | **100.0** / 97.0 | **100.0** / 96.5 | **100.0 / 98.0** | 99.5 / 97.0 |
| | Lcc | 52.5 / **51.6** | 36.7 / 37.2 | 47.1 / 51.1 | **54.0** / 51.5 | 52.1 / 51.1 | 52.7 / 50.7 | **52.8** / 51.5 |
| | RB-P | 48.3 / 56.2 | 48.3 / 54.7 | 47.4 / 54.9 | 48.1 / 56.6 | **50.0 / 57.1** | 47.4 / 57.0 | **50.1 / 58.2** |
| | Average | 44.4 / 42.2 | 46.4 / 40.5 | 40.3 / 37.8 | 45.3 / 42.1 | **48.9 / 45.6** | 45.5 / 41.6 | **48.9** / 46.4 |
| | NrtvQA | 23.1 / **18.7** | 21.9 / 13.0 | 21.6 / 18.4 | 24.3 / 17.7 | **27.7** / 14.2 | 25.3 / **18.3** | **27.0** / 12.1 |
| | Qasper | 20.0 / 12.7 | 15.9 / 6.0 | 18.4 / 13.1 | 21.1 / 13.6 | **28.5 / 26.2** | 21.6 / 15.2 | **28.6 / 26.8** |
| | MF-en | 23.6 / 26.2 | 29.2 / 25.0 | 23.0 / 23.1 | 23.4 / 30.3 | **32.6 / 35.0** | 26.2 / 32.0 | **30.2 / 35.2** |
| | HotpotQA | 43.1 / 37.3 | 40.1 / 30.9 | 37.2 / 31.2 | 44.5 / **33.5** | **47.4** / 30.7 | **46.9 / 38.1** | 46.1 / 34.8 |
| | 2WikiMQA | 22.4 / 21.9 | 20.1 / **27.8** | 22.4 / 20.1 | 24.6 / 23.1 | **31.8** / 25.4 | 27.2 / 24.0 | **33.6 / 31.7** |
| | Musique | 16.9 / 16.9 | 12.8 / 13.2 | 14.0 / 13.9 | 20.0 / **18.3** | **20.7** / 13.4 | 18.1 / 17.3 | **22.3 / 18.9** |
| | Gov Report | 25.0 / 25.5 | 26.0 / **26.0** | 24.8 / 25.2 | 25.3 / 25.4 | **26.9** / 23.3 | 25.3 / 25.3 | **26.9 / 26.2** |
| | QMSum | 19.5 / 19.8 | 21.0 / 18.9 | 19.1 / 19.8 | 19.7 / 20.2 | **22.6 / 21.8** | 20.9 / 20.4 | **22.2 / 21.3** |
| 90% | Multi News | 17.1 / 17.9 | 20.7 / 19.8 | 20.1 / 20.1 | 20.8 / 20.8 | **23.2 / 21.6** | 20.6 / 20.7 | **23.0 / 21.5** |
| | TREC | 11.0 / 15.8 | **47.5** / 26.5 | 28.0 / 35.0 | 33.5 / **37.8** | 22.5 / 27.3 | **33.0 / 44.0** | 19.5 / 33.0 |
| | TriviaQA | 84.8 / 87.0 | 68.4 / 85.4 | 90.7 / 52.4 | 82.9 / **87.9** | **91.5** / 73.5 | 82.4 / **87.6** | **92.3** / 76.3 |
| | SAMSum | 31.6 / 29.7 | 28.7 / 38.9 | 34.4 / 31.8 | **38.5** / 28.8 | 37.5 / **40.9** | **39.3** / 30.4 | 33.7 / **38.7** |
| | Pcount | 5.0 / 3.6 | **12.0** / 2.6 | 4.0 / **4.0** | 6.0 / 3.5 | 5.5 / 3.3 | **8.0** / 3.9 | 7.9 / 2.4 |
| | Pre | 49.5 / 42.0 | 24.0 / 10.0 | 16.0 / 15.5 | 55.0 / **53.0** | **61.5** / 15.5 | 56.5 / **67.5** | 55.5 / 23.0 |
| | Lcc | 50.6 / 51.6 | 23.1 / 22.2 | 52.3 / 50.6 | **51.0 / 52.6** | 39.4 / 39.7 | **49.2 / 53.0** | 39.0 / 41.1 |
| | RB-P | 49.3 / 54.9 | 49.7 / **55.1** | 52.9 / 53.3 | 49.2 / 54.8 | 52.2 / 53.6 | 47.3 / **55.0** | **54.2** / 54.6 |
| | Average | 30.8 / 30.1 | 28.8 / 26.3 | 29.9 / 26.7 | 33.7 / **32.6** | **35.7** / 29.1 | 34.2 / **33.2** | **35.1** / 31.1 |

On LLaMA-8B, CurDKV outperforms SnapKV by +3.6% at 30% compression (48.9% vs. 45.3%) and by +2.0% points at 90% (35.7% vs. 33.7%). On Mistral-7B, CurDKV shows a similar trend, surpassing SnapKV by +2.9% at 30% (45.6% vs. 42.7%) and by +0.5% at 90% (33.2% vs. 32.7%). These gains are robust across task types, including multi-hop QA, summarization, and code completion. Norm-based heuristics such as Knorm and ChunkKV fail to consistently preserve semantic fidelity, often underperforming even under moderate compression. This is most evident in Mistral-7B's few-shot learning tasks (e.g., TriviaQA, SAMSum), where CurDKV maintains over 89% performance, other baselines trail by 5–10%.

We compare AdaCurDKV, our adaptive compression strategy with its static variant (CurDKV) and the attention-based adaptive baseline, AdaSnapKV. Across models and retention levels, AdaCurDKV consistently outperforms AdaSnapKV, while performing competitively with CurDKV. With LLaMA-8B under 30% KV compression, AdaCurDKV achieves the highest average score (49.1%), outperforming AdaSnapKV (45.5%) by +3.6% and CurDKV (48.9%) by +0.02%. At 90% retention, AdaCurDKV maintains a strong average score of 35.1%, closely trailing CurDKV (35.7%) and surpassing AdaSnapKV (34.2%). For Mistral-7B, AdaCurDKV achieves 45.2% performance at 30% retention, outperforming AdaSnapKV (42.1%), and remaining close to CurDKV (45.6%). Although AdaCurDKV's average score drops to 29.1% at 90% retention, it remains competitive and stable across all task types. While CurDKV typically attains the best or second-best overall score, AdaCurDKV provides additional robustness by allocating head-wise budgets proportional to leverage score mass, leading to improved performance on head-sensitive tasks, especially in summarization and multi-hop QA.

We conduct ablations (c.f. Figure 3) to assess two core design choices in `CurDKV`: the source of leverage scores and the use of random projections. At 30% compression, different leverage modules (key, value, key-value product) perform similarly, but at 90%, value-centric and combined variants show better robustness than key-only. Random projections offer marginal gains overall, with slightly improved stability at high compression (more elaborated discussion in Appendix C).

As shown in Table 3a–3b, `CurDKV` consistently demonstrates superior robustness across compression levels and model scales. For the Qwen-14B model [Yang et al., 2024], it performs competitively with SnapKV at 30% compression and achieves a clear margin at 90% compression, outperforming all baselines by over 4–6% on average. The trend becomes more pronounced in the larger Qwen-32B model, where `CurDKV` surpasses other adaptive and non-adaptive methods by a significant margin under both 30% and 90% compression. These results highlight that value-aware KV compression scales favorably with model size and remains effective even under extreme memory constraints.

### 4.3 Results on Ruler

We further evaluate `CurDKV` and `AdaCurDKV` on the Ruler benchmark, focusing on needle-in-a-haystack (NIAH)

Table 3: LongBench evaluation results for Qwen models at 30% and 90% compression.

(a) **Qwen-14B**

| ▼ | Task | Chunk | KNorm | Snap | Streaming | CurDKV |
|---|------|-------|-------|------|-----------|--------|
| | 2WikiMQA | 28.71 | 28.76 | 28.45 | 25.06 | **27.97** |
| | HotpotQA | 33.53 | 31.44 | **34.56** | 26.67 | 31.58 |
| 30% | MF-en | 30.41 | 29.27 | **32.11** | 23.33 | 31.58 |
| | Qasper | 24.84 | 21.89 | 25.38 | 22.08 | **27.04** |
| | Average | 29.37 | 27.34 | **30.12** | 22.91 | 29.63 |
| | 2WikiMQA | 12.39 | 10.04 | 13.37 | 9.56 | **20.88** |
| | HotpotQA | 21.09 | 9.73 | 21.34 | 14.51 | **21.54** |
| 90% | MF-en | 17.03 | 7.20 | 18.25 | 14.66 | **22.08** |
| | Qasper | 9.25 | 7.20 | 9.56 | 10.22 | **16.23** |
| | Average | 14.94 | 10.00 | 15.63 | 12.24 | **20.18** |

(b) **Qwen-32B**

| ▼ | Task | Chunk | KNorm | Snap | Streaming | CurDKV |
|---|------|-------|-------|------|-----------|--------|
| | 2WikiMQA | 51.64 | 40.14 | 53.85 | 34.10 | **58.07** |
| | HotpotQA | 58.64 | 46.94 | 57.52 | 47.15 | **62.60** |
| 30% | MF-en | 45.41 | 43.39 | **48.56** | 32.61 | 50.42 |
| | Qasper | 42.46 | 37.73 | 41.86 | 43.11 | **47.75** |
| | Average | 49.54 | 42.05 | 50.45 | 39.24 | **54.71** |
| | 2WikiMQA | 24.24 | 6.14 | 9.84 | 18.54 | **40.59** |
| | HotpotQA | 37.85 | 3.80 | 36.29 | 30.38 | **43.51** |
| 90% | MF-en | 24.31 | 13.12 | 23.04 | 22.03 | **31.24** |
| | Qasper | 13.62 | 7.02 | 14.29 | 14.56 | **16.10** |
| | Average | 25.00 | 7.52 | 24.10 | 21.38 | **32.86** |

subtasks, which measure a model's ability to recover rare, high-salience facts embedded in long distractor contexts. These tasks are particularly sensitive to the retention of semantically critical key-value tokens. Full cache results are highlighted in Table 4.

As shown in Table 5, `CurDKV` and `AdaCurDKV` achieve the highest average score (98.7% and 97.7%) under 30% compression, outperforming all baselines. `CurDKV` substantially outperform norm-based (ChunkKV: 87.0%, Knorm: 71.8%) and attention-based approaches (SnapKV: 80.4%, StreamingLLM: 68.2%). On particularly hard retrieval subtasks like MK-3 and MQ, `AdaCurDKV` scores 93.6% and 100.0% respectively, indicating its strong capacity to retain tokens that encode key factual content. In S-3 – a shallow, multi-hop setting, `AdaCurDKV` scores 88.2% compared to 82.4% (AdaSnapKV) and 94.1% (`CurDKV`), reflecting robustness in both adaptive and static modes.

Table 4: Results of LLaMA-8B and Mistral-7B with full KV cache on Ruler (Needle-in-a-haystack subtasks).

| SubTask | LLaMA-8B | Mistral-7B |
|---------|----------|------------|
| S-1 | 100.0 | 95.2 |
| S-2 | 100.0 | 98.3 |
| S-3 | 100.0 | 100.0 |
| MK-1 | 100.0 | 97.9 |
| MK-2 | 100.0 | 97.8 |
| MK-3 | 97.9 | 80.9 |
| MQ | 99.5 | 85.6 |
| MV | 100.0 | 88.8 |
| Average | 99.6 | 92.8 |

At 90% compression, performance drops sharply for most baselines, with norm-based methods like Knorm (13.9%) and StreamingLLM (10.1%) failing to preserve fidelity. In contrast, `CurDKV` (34.7%) and `AdaCurDKV` (39.1%) maintain significantly higher performance, with `AdaCurDKV` again achieving the best overall average. Notably, on MQ and MV, `AdaCurDKV` reaches 56.7% and 52.0% respectively, far ahead of all other methods, highlighting its advantage on tasks requiring deeper contextual aggregation. On S-2, `AdaCurDKV` outperforms both AdaSnapKV and `CurDKV` (63.2% vs. 36.8% and 57.9%), showcasing the benefit of adaptive head-wise allocation in early-token regimes. Albeit the strong performance across different subtasks, we observe marginal underperformance on a subset of NIAH tasks with Mistral at 90% compression ratio. This can be attributed to Mistral's use of sliding-window attention which reduces head specialization and makes head-local token selection less effective. In such cases, heuristics like ChunkKV, which retain tokens based on uniform position or locality, can incidentally preserve critical information. Nonetheless, `CurDKV` continues to show superior performance on more challenging retrieval subtasks, indicating its robustness under complex attention patterns.

Table 5: KV compression methods on needle-in-a-haystack tasks for Llama-3.1-8B-Instruct and Mistral-7B-Instruct (▼: compression ratio). Friedman statistic of 25.6/21.7 (for 30% compression) and 12.3 (for 90% compression) with p-values $< 0.05$ indicate the statistical significance of the result obtained by `CurDKV` over the baselines. Full results for other compression ratios are highlighted in Tables 16 and 17 of Appendix C. Each result cell "$x/y$" indicate accuracies with LLaMA/Mistral.

| ▼ | Task | ChunkKV | Knorm | Streaming LLM | SnapKV | CurDKV | AdaSnapKV | AdaCurDKV |
|---|---|---|---|---|---|---|---|---|
| | | | | Non-Adaptive Methods | | | Adaptive Methods | |
| | S-1 | **100.0** / 93.6 | **100.0 / 96.8** | 62.9 / 61.3 | **100.0** / 54.8 | **100.0** / 74.2 | **100.0** / 64.5 | **100.0 / 85.5** |
| | S-2 | **100.0** / 91.2 | **100.0** / 12.3 | 64.9 / 64.9 | **100.0** / 63.2 | **100.0** / 91.2 | **100.0 / 80.7** | **100.0 / 96.5** |
| | S-3 | **96.1** / 56.9 | 51.0 / 0.0 | 78.4 / **78.4** | 25.5 / 2.0 | 94.1 / 72.6 | 82.4 / 13.7 | **88.2 / 60.8** |
| | MK-1 | **100.0** / 89.6 | 91.7 / 0.0 | 70.8 / 64.6 | **100.0** / 41.7 | **100.0** / 91.7 | **100.0 / 77.1** | **100.0 / 93.8** |
| 30% | MK-2 | 57.8 / 57.8 | 20.0 / 0.0 | 71.1 / 68.9 | 71.1 / 44.4 | **100.0 / 93.3** | 97.8 / **84.4** | **100.0** / 84.4 |
| | MK-3 | 44.7 / **46.8** | 17.0 / 2.1 | 55.3 / 31.9 | 53.2 / 29.8 | **95.7** / 23.4 | 89.4 / **68.1** | 93.6 / 44.7 |
| | MQ | **100.0 / 85.1** | 95.7 / 1.9 | 74.0 / 70.7 | 99.5 / 31.7 | 99.5 / 82.2 | **100.0** / 58.7 | **100.0 / 83.7** |
| | MV | 97.4 / 88.8 | 98.7 / 1.3 | 68.4 / 68.4 | 94.1 / 31.6 | **100.0 / 93.4** | **100.0** / 64.5 | **100.0 / 95.4** |
| | Average | 87.0 / 76.2 | 71.8 / 14.3 | 68.2 / 63.6 | 80.4 / 37.4 | **98.7 / 77.8** | 96.2 / 64.0 | 97.7 / 80.6 |
| | S-1 | **100.0 / 88.7** | **100.0** / 46.8 | 6.5 / 6.5 | 72.6 / 35.5 | 91.9 / 30.7 | 85.5 / **46.8** | **96.8** / 16.1 |
| | S-2 | 38.6 / 3.5 | 1.8 / 0.0 | 14.0 / **12.3** | 38.6 / 3.5 | **57.9** / 5.3 | 36.8 / 3.5 | **63.2** / **5.3** |
| | S-3 | **17.7** / 2.0 | 0.0 / 0.0 | 11.8 / **11.8** | 2.0 / 2.0 | 0.0 / 0.0 | **2.0 / 2.0** | 0.0 / 0.0 |
| | MK-1 | 29.2 / **12.5** | 0.0 / 0.0 | 10.4 / 8.3 | 14.6 / 4.2 | **39.6** / 8.3 | 20.8 / **6.3** | **39.6** / **6.3** |
| 90% | MK-2 | **8.9** / 6.7 | 0.0 / 0.0 | **8.9 / 8.9** | **8.9** / 0.0 | 4.4 / 0.0 | **11.1 / 2.2** | 4.4 / 0.0 |
| | MK-3 | 8.5 / **8.5** | 2.1 / 0.0 | **12.8** / 0.0 | 2.1 / 0.0 | 0.0 / 0.0 | **4.3 / 2.1** | 0.0 / 0.0 |
| | MQ | 28.4 / **13.9** | 4.3 / 0.0 | 9.6 / 6.7 | 15.4 / 11.5 | **38.5** / 11.1 | 14.9 / **11.5** | **56.7** / 1.4 |
| | MV | 30.9 / **17.8** | 3.3 / 0.0 | 7.2 / 7.2 | 21.7 / 15.8 | **45.4** / 12.5 | 19.7 / **15.8** | **52.0** / 9.9 |
| | Average | 32.8 / **19.2** | 13.9 / 5.8 | 10.1 / 7.7 | 22.0 / 9.1 | **34.7** / 7.8 | 24.4 / **11.3** | **39.1** / 3.9 |

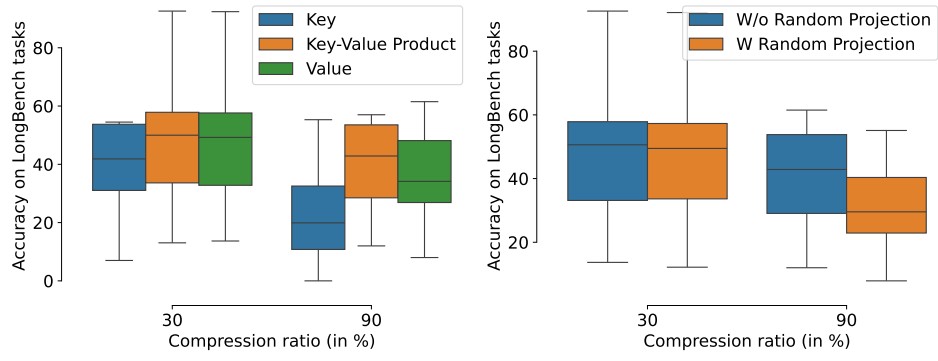

(a) Leverage computation type  (b) Ablation with and without random projection

Figure 3: Accuracy of LLaMA-3.1-8B-Instruct on LongBench tasks for different ablations of `CurDKV`. Full results reported in Table 18 in Appendix C.

## 4.4 Computational Efficiency of `CurDKV`

To evaluate the practical benefits of KV compression beyond accuracy, we analyze `CurDKV` in terms of memory usage, prefill latency, and generation latency across varying sequence lengths and compression ratios. As expected, `CurDKV` yields (c.f. Figure 4a) a near-linear reduction in KV cache size with increasing compression ratio. For example, at 80% compression, memory usage drops from 15.6 GB to under 3 GB for a 128K-token context. This linear trend holds consistently across all tested sequence lengths. Since `CurDKV` compresses the cache on a per-layer and per-head basis, these

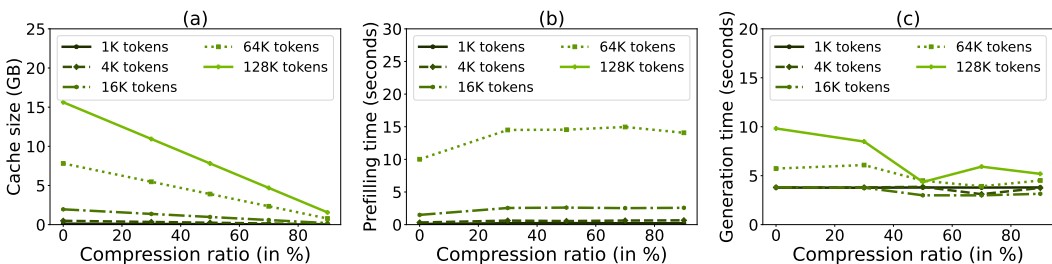

Figure 4: Prefilling and generation statistics of `CurDKV` for different text lengths with LLaMA-8B.

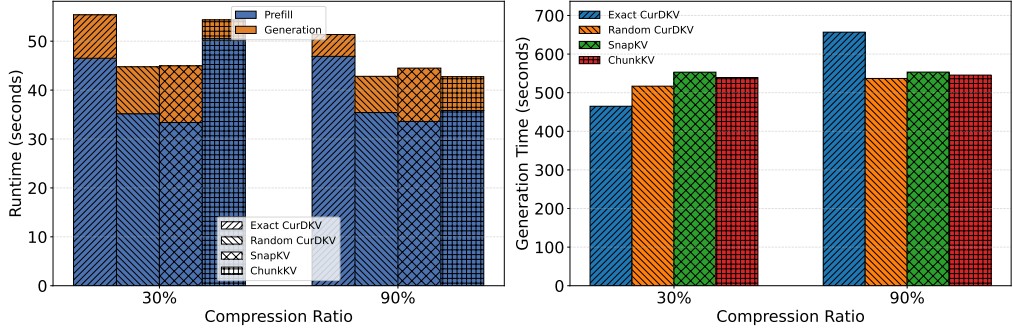

(a) Breakdown analysis of runtime of LLaMA-8B model with generation length of 100.

(b) Comparative analysis of generation runtime with 4K generation length.

Figure 5: Comparative analysis of different compression methods with 128K context length.

reductions are directly proportional to the retained key-value tokens, enabling predictable memory scaling in long-context scenarios. Figure 4(b) shows a slight upward trend in prefill latency at higher compression ratios, especially for longer sequences. This is because CurDKV computes leverage scores and performs token selection during the prefill phase, introducing modest computational overhead. For instance, with 128K tokens, prefill time increases from approximately 10 seconds (no compression) to around 14–15 seconds at high compression levels. However, the added cost remains constant beyond 40–60% compression, indicating amortization of overhead due to batch-efficient scoring. Interestingly, generation latency (Figure 4(c)) decreases with compression for all sequence lengths, particularly for longer inputs. This is attributed to the smaller number of cached tokens being read during autoregressive decoding. With 128K-token contexts, generation time reduces from 10 seconds (no compression) to under 6 seconds at 80% compression. The effect is more muted for shorter sequences, but consistently observable. These properties make CurDKV a practical and scalable solution for deployment-time KV cache optimization in LLMs.

As shown in the runtime and generation analyses in Figure 5, CurDKV achieves a favorable balance between latency and accuracy across compression levels. At 30% compression, CurDKV with random projection yields the lowest overall runtime due to reduced prefill cost, while maintaining comparable generation latency to other baselines. Under 90% compression, it continues to outperform in total inference time, benefiting from efficient subspace projection of key–value pairs. The longer-sequence generation study further confirms these trends, where CurDKV consistently achieves faster decoding than SnapKV and ChunkKV, with up to 15% reduction in generation time. At higher compression, exact CurDKV (without random projection) yields higher generation time, due to iterative CUR decomposition; however, CurDKV with random projection achieves the least generation time, highlighting the trade-offs between exact and randomized CurDKV.

## 5 Conclusion

In this paper we proposed CurDKV, a value-centric KV cache compression method based on CUR decomposition, and AdaCurDKV, its adaptive variant. Unlike prior approaches that rely on attention scores, our method selects key-value tokens using leverage scores derived from the value matrix, with efficient approximations via random projections. Experiments on LongBench and Ruler benchmarks showed that CurDKV consistently outperformed existing methods across tasks and compression ratios, while remaining compatible with FlashAttention and Grouped Query Attention.

**Limitations and future work.** While CurDKV offers a principled and effective approach to KV cache compression, it relies on static token selection during the prefill phase, which may limit its responsiveness to dynamically emerging information needs during generation. Future work could explore query-aware token selection, hybrid token-chunk strategies, or lightweight learned components to improve performance in semantically sparse settings. Extending CurDKV to support dynamic compression during generation is another promising direction to better adapt to evolving query demands.

## Acknowledgments

T. Chakraborty acknowledges the support of the IBM-IITD AI Horizons network and Rajiv Khemani Young Faculty Chair Professorship in Artificial Intelligence. He acknowledges the support of Google GCP Grant for providing the necessary computational resources.

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

# A Proof of Lemma 3.1

*Proof.* Let $A = \text{softmax}(QK^T)V$, and let $A' = \text{softmax}(QK'^T)V'$. In this proof, we will upper bound $\|A - A'\|_F$. Throughout the proof, we also assume that $K$ and $K'$ have the same dimensions, which is achieved by adding zero rows to $K'$ as necessary (and the same holds for $V$ and $V'$).

First, we have

$$A - A' = \text{softmax}(QK^T)V - \text{softmax}(QK^T)V' + \text{softmax}(QK^T)V' - \text{softmax}(QK'^T)V'$$
$$= \text{softmax}(QK^T)(V - V') + (\text{softmax}(QK^T) - \text{softmax}(QK'^T))V'$$

and hence using the triangle inequality for norms, we get

$$\|A - A'\|_F \leq \|\text{softmax}(QK^T)(V - V')\|_F + \|(\text{softmax}(QK^T) - \text{softmax}(QK'^T))V'\|_F$$

Next, if $\|\cdot\|_{\text{op}}$ is the operator norm, then we have that $\|AB\|_F \leq \|A\|_{\text{op}}\|B\|_F$. Applying this to the two terms on the RHS above, we get

$$\|A - A'\|_F \leq \|\text{softmax}(QK^T)\|_{\text{op}}\|V - V'\|_F + \|\text{softmax}(QK^T) - \text{softmax}(QK'^T)\|_{\text{op}}\|V'\|_F$$
$$\leq \|\text{softmax}(QK^T)\|_{\text{op}}\|V - V'\|_F + (\|\text{softmax}(QK^T)\|_{\text{op}} + \|\text{softmax}(QK'^T)\|_{\text{op}})\|V'\|_F$$

where in the second step above, we have again used the triangle inequality for norms. Now, we know that both $\text{softmax}(QK^T)$ and $\text{softmax}(QK'^T)$ are row-stochastic matrices; hence, the $\|\cdot\|_{\text{op}}$ norm of both these matrices is exactly $\sqrt{n}$. So, we get that

$$\|A - A'\|_F \leq \sqrt{n}\|V - V'\|_F + 2\sqrt{n}\|V'\|_F$$

completing the proof of the claim.

$\square$

# B Datasets, Task Descriptions and Task Templates

## B.1 The LongBench benchmark

LongBench [Bai et al., 2023] is a set of multitasks designed to assess the ability of LLMs to handle long-context problems requiring deep understanding and reasoning. Tasks include single-document question-answering, mutli-document question answering, summarization, few-shot learning, synthetic tasks and code generation. Below we present a detailed overview of all tasks:

- **Single-Doc QA:** Models are required to obtain the answer from a single source document. Test samples are derived from a variety of datasets, including NarrativeQA [Kočiský et al., 2017], Qasper [Dasigi et al., 2021], and MultiFieldQA [Liu et al., 2023a]. Templates for these tasks can be found in Table 6.
- **Multi-Doc QA:** Models are required to extract and combine information from multiple source documents to obtain the answer. Data samples are derived from three multi-hop QA datasets: HotpotQA [Yang et al., 2018], 2WikiMultihopQA [Ho et al., 2020] and MuSiQue [Trivedi et al., 2022]. Templates for these tasks can be found in Table 7.
- **Summarization:** These tasks require a comprehensive understanding of the context. The data samples are obtained from the GovReport [Huang et al., 2021] and QMSum [Zhong et al., 2021] datasets. Templates for these tasks can be found in Table 8.
- **Few-shot Learning:** These tasks have classification, summarization and reading-comprehension tasks integrated within them to maintain task diversity. The TREC dataset [Li and Roth, 2002] is used for classification problems. The SAMSum dataset [Gliwa et al., 2019] is used for summarization tasks. Finally, TriviaQA [Joshi et al., 2017] is utilized for comprehension tasks. Templates for these tasks can be found in Table 9.
- **Synthetic Tasks:** These tasks assess a model's ability to handle specific scenarios and patterns. In LongBench, the PassageRetrieval-en and PassageCount datasets are used. Templates for these tasks can be found in Table 10.
- **Code Completion Tasks:** These tasks are meant to assist users by completing code based on a user's input and context. The LCC dataset from the original Long Code Completion dataset [Guo et al., 2023] as well as the RepoBench-P dataset [Liu et al., 2023b] is used. Templates for these tasks can be found in Table 11.

Table 6: LongBench templates for Single-Doc QA tasks.

| | |
|---|---|
| NarrativeQA | **Task Template:**
You are given a story, which can be either a novel or a movie script, and a question. Answer the question as concisely as you can, using a single phrase if possible. Do not provide any explanation.

Story: {context}

Now, answer the question based on the story as concisely as you can, using a single phrase if possible. Do not provide any explanation.

Question: {question} |
| Qasper | **Task Template:**
You are given a scientific article and a question. Answer the question as concisely as you can, using a single phrase or sentence if possible. If the question cannot be answered based on the information in the article, write "unanswerable". If the question is a yes/no question, answer "yes", "no" or "unanswerable". Do not provide any explanation.

Article: {context}

Answer the question based on the above article as concisely as you can, using a single phrase or sentence if possible. If the question cannot be answered based on the information in the article, write "unanswerable". If the question is a yes/no question, answer "yes", "no" or "unanswerable". Do not provide any explanation

Question: {question} |
| MultifieldQA EN | **Task Template:** Read the following text and answer briefly.

{context}

Now, answer the following question based on the above text, only give me the answer and do not output any other words.

Question: {question} |

## B.2 The Ruler benchmark

Ruler [Hsieh et al., 2024] is a collection of synthetic examples to evaluate long-context language models, containing four task categories to test behaviours beyond simple retrieval from context. Here is a detailed description of each task category:

- **Singe Needle-In-A-Haystack (S-):** In these tasks, a keyword sentence, called the "needle", is embedded within a lengthy text, called the "haystack". The objective of the task is to retrieve the needle from the context.
- **Multi-keys NIAH (MK-):** These tasks are similar to **S-** tasks, with the difference being the presence of multiple "needles" inserted into the "haystack". However, the task is to retrieve only one of them.
- **Multi-values NIAH (MV-):** Here, multiple "needles" in the form of key-value pairs are hidden in the "haystack". The task is to retrieve all values associated to a single key.
- **Multi-queries NIAH (MQ-):** Here, multiple "needles" in the form of key-value pairs are inserted into the "haystack". All "needles" corresponding to the keys specified in the queries are to be retrieved.

Templates for these tasks can be found in Tables 12 and 13 respectively.

Table 7: LongBench templates for Multi-Doc QA tasks.

| | |
|---|---|
| HotpotQA | **Task Template:**
Answer the question based on the given passages. Only give me the answer and do not output any other words.

The following are given passages.
{context}

Answer the question based on the given passages. Only give me the answer and do not output any other words.

Question: {question} |
| 2WikimQA | **Task Template:**
Answer the question based on the given passages. Only give me the answer and do not output any other words.

The following are given passages.
{context}

Answer the question based on the given passages. Only give me the answer and do not output any other words.

Question: {question} |
| Musique | **Task Template:**
Answer the question based on the given passages. Only give me the answer and do not output any other words.

The following are given passages.
{context}

Answer the question based on the given passages. Only give me the answer and do not output any other words.

Question: {question} |

# C   Additional Results

## C.1   Results on LongBench

We evaluate the performance of `CurDKV` and `AdaCurDKV` against baseline compression methods at intermediate compression ratios of 50% and 70%, using both the LLaMA-3.1-8B and Mistral-7B in Table 14 and Table 15, respectively.

On LLaMA, `CurDKV` achieves the highest average accuracy (48.3), outperforming all non-adaptive baselines such as SnapKV (44.3) and Knorm (43.1), as well as Streaming LLM (37.7). The adaptive variant `AdaCurDKV` also performs competitively, with an average score of 47.9, surpassing AdaSnapKV (44.7). At 70% compression, CUR-based approaches continue to perform favorably. On LLaMA, `CurDKV` again achieves the highest average (45.4), ahead of SnapKV (41.9) and Knorm (36.4), confirming its robustness across moderate compression levels. `AdaCurDKV` further improves upon AdaSnapKV (42.4) with an average of 46.1, marking consistent gains across adaptive strategies.

On Mistral, `CurDKV` again delivers the best average score (43.8), outperforming SnapKV (41.2) and ChunkKV (41.0). `AdaCurDKV` (43.7) also surpasses AdaSnapKV (41.9), particularly benefiting from adaptive budget reallocation on heterogeneous tasks such as TREC (72.0) and SAMSum (45.4). Across both models, CUR-based methods show superior stability, particularly under structured reasoning and information-dense settings.

Table 8: LongBench templates for Summarization tasks.

| | |
|---|---|
| Gov Report | **Task Template:**
You are given a report by a government agency. Write a one-page summary of the report.

Report:
{context}

Now, write a one-page summary of the report. |
| QMSum | **Task Template:**
You are given a meeting transcript and a query containing a question or instruction. Answer the query in one or more sentences.

Transcript
{context}

Now, answer the query based on the above meeting transcript in one or more sentences.

Query: {question} |
| Multi News | **Task Template:**
You are given several news passages. Write a one-page summary of all news.

News
{context}

Now, write a one-page summary of all the news. |

Table 9: LongBench templates for Few-shot learning tasks.

| | |
|---|---|
| TREC | **Task Template:**
Please determine the type of the question below. Here are some examples of questions.

{context}
{question} |
| TriviaQA | **Task Template:**
Answer the question based on the given passage. Only give me the answer and do not output any other words. The following are some examples.

{context}
{question} |
| SAMSum | **Task Template:**
Summarize the dialogue into a few short sentences. The following are some examples.

{context}
{question} |

Table 10: LongBench templates for Synthetic tasks.

| | |
|---|---|
| Passage Count | **Task Template:**
There are some paragraphs below sourced from Wikipedia. Some of them may be duplicates. Please carefully read these paragraphs and determine how many unique paragraphs there are after removing duplicates. In other words, how many non-repeating paragraphs are there in total?

{context}

Please enter the final count of unique paragraphs after removing duplicates. The output format should only contain the number, such as 1, 2, 3, and so on. |
| Passage Retrieval EN | **Task Template:**
Here are 30 paragraphs from Wikipedia, along with an abstract. Please determine which paragraph the abstract is from.

{context}

The following is an abstract.

{question}

Please enter the number of the paragraph that the abstract is from. The answer format must be like "Paragraph 1", "Paragraph 2", etc. |

Table 11: LongBench templates for Code tasks.

| | |
|---|---|
| Lcc | **Task Template:**
Please complete the code given below.
{context}
Next line of code: |
| Repobench-P | **Task Template:**
Please complete the code given below.
{context}
{question}
Next line of code: |

## C.2 Results on Ruler

Table 16 and 17 report the results of different KV compression models with LLaMA and Mistral models, respectively, at 50% and 70% compression ratios.

With LLaMA, at 50% compression, `CurDKV` achieves a strong average of 83.1, outperforming all non-adaptive baselines, including ChunkKV (79.2) and SnapKV (67.0). The adaptive variant, `AdaCurDKV` further improves performance to 89.6, benefiting from its dynamic budget allocation across heads. Notably, `AdaCurDKV`achieves perfect accuracy (100.0) on 6 out of 8 subtasks, demonstrating its ability to preserve high-salience tokens even at moderate compression. At 70% compression, the relative advantage of CUR-based methods persists. `CurDKV` scores an average of 62.0, substantially ahead of SnapKV (48.0) and Knorm (32.7), however, falling short from the best non-adaptive baseline ChunkKV (64.1). `AdaCurDKV` continues to perform best overall with an average of 69.8, highlighting its robustness in token-starved regimes. These results reinforce the effectiveness of CUR-based compression for preserving semantically critical tokens required for pinpoint retrieval.

On Mistral at 50% compression, `CurDKV` achieves an average score of 54.2, considerably higher than SnapKV (23.9), Knorm (12.0), and Streaming LLM (41.3). `AdaCurDKV` pushes this further to 56.1, showing consistent gains across nearly all subtasks.

As shown in Tables 19, even for a longer context length of 128K, `CurDKV` achieves the strongest overall performance on the NIAH benchmark, particularly under higher compression. At 30% compression, it attains the highest average accuracy (79.95%), marginally surpassing SnapKV while maintaining consistent gains across multi-key and multi-value retrieval tasks. Under 50% compression, the

Table 12: Ruler templates for S and MK tasks.

| | |
|---|---|
| S-1 | **Task Template:**
Some special magic numbers are hidden within the following text. Make sure to memorize it. I will quiz you about the numbers afterwards.
The grass is green. The sky is blue. The sun is yellow. Here we go. There and back again.
...... One of the special magic numbers for {word} is: {number}. ......
What is the special magic number for {word} mentioned in the provided text?

The special magic number for {word} mentioned in the provided text is |
| S-2 | **Task Template:**
Some special magic numbers are hidden within the following text. Make sure to memorize it. I will quiz you about the numbers afterwards.
Paul Graham Essays.
...... One of the special magic numbers for {word} is: {number}. ......
What is the special magic number for {word} mentioned in the provided text?

The special magic number for {word} mentioned in the provided text is |
| S-3 | **Task Template:**
Some special magic numbers are hidden within the following text. Make sure to memorize it. I will quiz you about the numbers afterwards.
Paul Graham Essays.
...... One of the special magic numbers for {word} is: {number}. ......
What is the special magic number for {word} mentioned in the provided text?

The special magic number for {word} mentioned in the provided text is |
| MK-1 | **Task Template:**
Some special magic numbers are hidden within the following text. Make sure to memorize it. I will quiz you about the numbers afterwards.
Paul Graham Essays.
...... One of the special magic numbers for {word-1} is: {number-1}. ......
...... One of the special magic numbers for {word-2} is: {number-2}. ......
...... One of the special magic numbers for {word-3} is: {number-3}. ......
...... One of the special magic numbers for {word-4} is: {number-4}. ......
What is the special magic number for {word-4} mentioned in the provided text?

The special magic number for {word-4} mentioned in the provided text is |
| MK-2 | **Task Template:**
Some special magic numbers are hidden within the following text. Make sure to memorize it. I will quiz you about the numbers afterwards.
Paul Graham Essays.
...... One of the special magic numbers for {word-1} is: {number-1}. ......
...... One of the special magic numbers for {word-2} is: {number-2}. ......
...... One of the special magic numbers for {word-x} is: {number-x}. ......
...... One of the special magic numbers for {word-n-1} is: {number-n-1}. ......
...... One of the special magic numbers for {word-n} is: {number-n}. ......
What is the special magic number for {word-x} mentioned in the provided text?

The special magic number for {word-x} mentioned in the provided text is |
| MK-3 | **Task Template:**
Some special uuids are hidden within the following text. Make sure to memorize it. I will quiz you about the uuids afterwards.
Paul Graham Essays.
...... One of the special magic uuids for {uuid-1} is: {uuid-1}. ......
...... One of the special magic uuid for {uuid-2} is: {uuid-2}. ......
...... One of the special magic uuid for {uuid-x} is: {uuid-x}. ......
...... One of the special magic uuid for {uuid-n-1} is: {uuid-n-1}. ......
...... One of the special magic uuid for {uuid-n} is: {uuid-n}. ......
What is the special magic number for {uuid-x} mentioned in the provided text?

The special magic number for {uuid-x} mentioned in the provided text is |

Table 13: Ruler templates for MV and MQ tasks.

| | |
|---|---|
| MV | **Task Template:**
Some special magic numbers are hidden within the following text. Make sure to memorize it. I will quiz you about the numbers afterwards.
Paul Graham Essays.
...... One of the special magic numbers for {word} is: {number-1}. ......
...... One of the special magic numbers for {word} is: {number-2}. ......
...... One of the special magic numbers for {word} is: {number-3}. ......
...... One of the special magic numbers for {word} is: {number-4}. ......
What are all the special magic numbers for {word} mentioned in the provided text?

The special magic numbers for {word} mentioned in the provided text are |
| MQ | **Task Template:**
Some special magic numbers are hidden within the following text. Make sure to memorize it. I will quiz you about the numbers afterwards.
Paul Graham Essays.
...... One of the special magic numbers for {word=1} is: {number-1}. ......
...... One of the special magic numbers for {word-2} is: {number-2}. ......
...... One of the special magic numbers for {word-3} is: {number-3}. ......
...... One of the special magic numbers for {word-4} is: {number-4}. ......
What are all the special magic numbers for {word-1}, {word-2}, {word-3}, and {word-4} mentioned in the provided text?

The special magic numbers for {word-1}, {word-2}, {word-3}, and {word-4} mentioned in the provided text are |

advantage becomes more pronounced, where `CurDKV` yields the top accuracy on most long-context string-matching tasks and sustains a clear lead in overall average (68.69%). These results indicate that value-aware KV compression enhances retrieval fidelity even when memory budgets are severely constrained, preserving precise information access across longer contexts.

Albeit the minor performance drops, these experiments confirm that CUR-based selection, especially when adaptively guided, provides a robust and effective mechanism for maintaining retrieval fidelity under tight memory constraints, outperforming both norm- and attention-based KV selection across two architectures and multiple compression levels.

### C.3 Ablation Study

We perform two ablation studies to investigate the key design decisions behind `CurDKV` : (1) the module on which leverage scores are computed, and (2) the effect of using randomized projections for score approximation. While in our algorithm we use combined leverage scores computed using both key and value vectors (line 5 of Algorithm 1), in our ablations we study the affect of using only the key or value vectors to compute leverage scores. We also run experiments wherein leverage scores are computed on the original key/value matrices without the use of the Gaussian projection (*i.e.*, we drop lines 3-4 of Algorithm 1)

We first ablate the module on which leverage scores are computed. Figure 3 shows the average performance (task-level scores reported in Table 18) when using key-only, value-only, and the default key-value product (elementwise product of key and value leverage scores). All three variants perform similarly at 30% compression. However, at 90% compression, the differences become more pronounced: key-only scores degrade significantly, while value and key-value product maintain higher median accuracy and robustness. This confirms that value information is more predictive of output quality, and that combining key and value structure yields the most reliable token selection strategy.

Next, we evaluate the role of randomized leverage score approximation. This determines whether the key and value matrices are projected to a low-rank space via Gaussian random matrices before

Table 14: Results of KV compression methods on LongBench tasks with LLaMA-8B at 50% and 70% compression ratios (▼: compression ratio).

| ▼ | Task | ChunkKV | Knorm | Streaming LLM | SnapKV | CurDKV | AdaSnapKV | AdaCurDKV |
|---|------|---------|-------|---------------|--------|--------|-----------|-----------|
| | | | | **Non-Adaptive Methods** | | | **Adaptive Methods** | |
| | NrtvQA | 28.8 | 28.6 | 25.2 | 29.2 | **30.4** | 29.2 | **30.6** |
| | Qasper | 39.6 | 40.8 | 37.1 | 41.1 | **49.0** | 43.4 | **48.1** |
| | MF-en | 43.7 | 49.3 | 32.0 | 45.4 | **54.9** | 51.5 | **52.4** |
| | HotpotQA | 55.8 | 55.3 | 50.1 | **57.3** | 57.2 | **56.0** | 55.6 |
| | 2WikiMQA | 49.6 | 44.7 | 38.0 | **50.6** | 49.1 | **51.1** | 49.3 |
| | Musique | 27.9 | 26.4 | 23.7 | 30.7 | **34.7** | 30.0 | **32.0** |
| | Gov Report | 32.7 | 32.0 | 30.7 | 31.9 | **33.4** | 32.0 | **32.9** |
| | QMSum | 23.0 | 24.3 | 21.9 | 23.8 | **24.7** | 24.3 | **25.2** |
| 50% | Multi News | 25.6 | 25.4 | 25.6 | 25.4 | **26.5** | 25.9 | **26.6** |
| | TREC | 23.0 | 60.8 | 31.0 | 36.5 | **68.0** | 36.0 | **67.5** |
| | TriviaQA | 84.9 | 88.4 | 92.0 | 85.5 | **92.3** | 86.1 | **92.9** |
| | SAMSum | 39.8 | 34.3 | 37.5 | 40.0 | **40.7** | 40.5 | **41.2** |
| | Pcount | 10.6 | 10.6 | 7.5 | 11.1 | **12.8** | **11.1** | **11.1** |
| | Pre | 97.0 | 86.5 | 54.0 | **100.0** | 98.5 | **100.0** | 99.0 |
| | Lcc | 50.9 | 32.6 | 48.8 | **53.3** | 49.9 | **51.1** | 50.4 |
| | RB-P | 49.2 | 49.3 | 48.5 | 47.7 | **50.4** | 47.4 | **52.3** |
| | Average | 42.6 | 43.1 | 37.7 | 44.3 | **48.3** | 44.7 | **47.9** |
| | NrtvQA | **30.9** | 25.9 | 24.2 | 27.3 | 30.5 | 29.1 | **30.4** |
| | Qasper | 31.6 | 32.5 | 27.8 | 30.8 | **43.2** | 34.9 | **44.2** |
| | MF-en | 34.6 | 42.7 | 26.7 | 37.0 | **45.1** | 41.9 | **46.5** |
| | HotpotQA | 53.6 | 49.1 | 43.5 | **54.8** | 51.7 | **55.6** | 54.6 |
| | 2WikiMQA | 38.5 | 33.3 | 30.9 | **46.9** | 45.2 | **47.5** | 45.5 |
| | Musique | 25.8 | 19.7 | 20.8 | 27.5 | **29.3** | 29.9 | **30.7** |
| | Gov Report | 30.6 | 29.4 | 29.2 | 29.9 | **31.2** | 29.3 | **30.7** |
| | QMSum | 22.3 | 23.5 | 21.1 | 22.6 | **24.4** | 22.9 | **24.5** |
| 70% | Multi News | 23.5 | 23.8 | 24.2 | 23.8 | **26.0** | 24.4 | **25.9** |
| | TREC | 15.0 | 54.5 | 31.5 | 39.0 | **58.0** | 34.0 | **63.7** |
| | TriviaQA | 84.0 | 74.9 | 91.5 | 84.5 | **92.0** | 85.3 | **91.9** |
| | SAMSum | 36.3 | 27.7 | 36.3 | 41.5 | **41.7** | **41.8** | 41.2 |
| | Pcount | 7.6 | 9.1 | 6.3 | 10.1 | **10.8** | **10.1** | 9.7 |
| | Pre | 92.5 | 58.5 | 36.5 | 93.0 | **98.5** | 96.0 | **98.5** |
| | Lcc | 51.0 | 28.5 | 50.9 | **53.2** | 46.7 | **50.6** | 46.4 |
| | RB-P | 49.4 | 48.6 | 49.9 | 47.8 | **51.6** | 45.9 | **52.8** |
| | Average | 39.2 | 36.4 | 34.4 | 41.9 | **45.4** | 42.4 | **46.1** |

computing leverage scores. We observe marginal differences between the two variants. At 30% compression, both versions perform similarly in median accuracy, though random projection slightly improves the worst-case performance. At 90% compression, however, random projections result in modest improvements in median and lower-quartile accuracy. These gains suggest that random projections can help stabilize leverage score estimation under extreme compression, but their effect is secondary to the choice of leverage module.

We analyze the robustness of `AdaCurDKV` with respect to the safeguard parameter $\alpha$ and the Gaussian projection dimension $r$. As shown in Tables 20a and 20b, performance remains remarkably stable across $\alpha \in \{0.1, 0.5, 0.9\}$, indicating that the adaptive safeguard mechanism is largely self-regularizing. The average scores vary by less than one point even under aggressive 90% compression, demonstrating strong resilience. Increasing $r$ consistently improves performance, saturating around $r = 100$, suggesting that moderate projection ranks suffice for accurate subspace estimation without incurring high overhead. At the same time, increasing $r$ converges to the exact CUR-decomposition of key-value matrices.

## C.4 Runtime Analysis

We analyze the runtime behavior of `CurDKV` and SnapKV under varying compression ratios and input lengths, as illustrated in Figures 4 and 6. We focus on three key metrics: KV cache size, prefill time, and generation time.

**Cache Size.** Both methods exhibit nearly linear reduction in KV cache size as compression increases. For example, with a 128K token input, the cache size reduces from approximately 16 GB at 0% compression to under 2 GB at 80% compression for both `CurDKV` and SnapKV (Figures 4a and 6a).

Table 15: Results of KV compression methods on LongBench tasks with Mistral-7B at 50% and 70% compression ratios (▾: compression ratio).

| ▾ | Task | ChunkKV | Knorm | Streaming LLM | SnapKV | CurDKV | AdaSnapKV | AdaCurDKV |
|---|---|---|---|---|---|---|---|---|
| | | | | **Non-Adaptive Methods** | | | **Adaptive Methods** | |
| 50% | NrtvQA | 24.6 | 20.5 | 23.5 | 24.3 | **25.6** | 24.4 | **25.1** |
| | Qasper | 30.4 | 29.0 | 30.8 | 32.6 | **39.4** | 30.3 | **39.2** |
| | MF-en | 44.1 | 42.5 | 30.8 | 47.1 | **51.3** | 48.5 | **49.4** |
| | HotpotQA | 47.0 | 45.5 | 40.8 | **47.2** | 42.7 | **48.5** | 44.8 |
| | 2WikiMQA | 35.0 | 31.3 | 30.5 | 36.5 | **39.5** | 36.8 | **37.7** |
| | Musique | 25.1 | 22.5 | 18.6 | 23.0 | **27.7** | 26.0 | **26.9** |
| | Gov Report | 32.4 | 30.0 | 31.7 | 31.6 | **33.2** | **31.7** | 31.4 |
| | QMSum | 23.5 | 23.2 | 22.9 | 23.7 | **24.9** | 24.0 | **24.9** |
| | Multi News | 25.2 | 24.7 | 25.1 | 25.2 | **26.6** | 25.5 | **26.7** |
| | TREC | 48.8 | 40.7 | 56.0 | 50.0 | **69.0** | 51.0 | **72.0** |
| | TriviaQA | 86.8 | 86.9 | 69.9 | 86.5 | **87.4** | 87.7 | 85.2 |
| | SAMSum | 22.3 | 36.6 | 21.3 | 21.4 | **45.7** | 22.7 | **45.4** |
| | Pcount | **7.0** | 4.2 | 1.5 | 4.7 | 5.9 | 5.8 | **6.6** |
| | Pre | 96.0 | 67.5 | 53.5 | **96.5** | 78.0 | **97.5** | 68.5 |
| | Lcc | **52.4** | 32.8 | 52.4 | 52.2 | 47.8 | 52.1 | **57.4** |
| | RB-P | 55.5 | 54.3 | 53.9 | **56.8** | 55.3 | 57.3 | **58.7** |
| | Average | 41.0 | 37.0 | 35.2 | 41.2 | **43.8** | 41.9 | **43.7** |
| 70% | NrtvQA | 21.3 | 16.5 | 20.4 | **23.0** | 21.02 | **23.2** | 19.1 |
| | Qasper | 21.4 | 19.2 | 21.6 | 24.2 | **35.1** | 25.1 | **35.0** |
| | MF-en | 38.2 | 34.1 | 27.8 | 41.9 | **46.8** | 42.1 | **46.0** |
| | HotpotQA | **44.1** | 38.9 | 38.1 | 43.9 | 38.2 | 45.5 | 34.3 |
| | 2WikiMQA | **33.6** | 26.0 | 26.9 | 29.2 | 32.3 | 31.7 | **33.9** |
| | Musique | 20.9 | 18.9 | 16.7 | **21.2** | 18.0 | **21.9** | 15.3 |
| | Gov Report | **29.9** | 27.1 | 29.6 | 29.7 | 27.0 | **29.2** | 23.3 |
| | QMSum | 21.9 | 21.6 | 22.1 | 22.2 | **23.6** | 22.4 | **23.1** |
| | Multi News | 23.5 | 22.6 | 23.6 | 23.7 | **25.4** | 24.3 | **25.1** |
| | TREC | 38.3 | 36.0 | 46.0 | 46.0 | **65.5** | 48.8 | **66.0** |
| | TriviaQA | **87.6** | 86.3 | 62.8 | 86.8 | 59.4 | **87.0** | 52.4 |
| | SAMSum | 25.4 | 39.6 | 25.7 | 22.2 | **42.4** | 23.0 | **42.1** |
| | Pcount | 5.6 | 3.3 | 2.5 | **5.9** | 5.1 | 4.2 | **5.7** |
| | Pre | 91.0 | 38.3 | 36.0 | **92.8** | 25.5 | **94.5** | 11.0 |
| | Lcc | **54.1** | 29.5 | 52.2 | 53.54 | 43.6 | 53.0 | **55.3** |
| | RB-P | 55.3 | 55.0 | 53.1 | **56.0** | 52.2 | 56.5 | **57.0** |
| | Average | 38.2 | 32.0 | 31.6 | **38.9** | 35.1 | **39.5** | 34.0 |

Table 16: Results of all KV compression baselines on needle-in-a-haystack subtasks with LLaMA-8B at 50% and 70% compression ratios (▾: compression ratio).

| ▾ | SubTask | ChunkKV | Knorm | Streaming LLM | SnapKV | CurDKV | AdaSnapKV | AdaCurDKV |
|---|---|---|---|---|---|---|---|---|
| | | | | **Non-Adaptive Methods** | | | **Adaptive Methods** | |
| 50% | S-1 | **100.0** | 100.0 | 38.7 | **100.0** | **100.0** | 100.0 | **100.0** |
| | S-2 | **100.0** | 96.5 | 40.4 | 98.3 | **100.0** | 100.0 | **100.0** |
| | S-3 | **88.2** | 9.8 | 51.0 | 11.8 | 25.5 | **60.8** | 54.9 |
| | MK-1 | 95.8 | 70.8 | 56.3 | **100.0** | **100.0** | 100.0 | **100.0** |
| | MK-2 | 40.0 | 6.7 | 48.9 | 40.0 | **82.2** | 93.3 | **95.6** |
| | MK-3 | 29.8 | 12.8 | 36.2 | 19.2 | **61.7** | 46.8 | **68.1** |
| | MQ | 95.2 | 71.6 | 50.5 | 85.6 | **99.0** | 98.6 | 98.1 |
| | MV | 84.2 | 80.3 | 48.0 | 80.9 | **96.7** | 100.0 | **100.0** |
| | Average | 79.2 | 56.1 | 46.2 | 67.0 | **83.1** | 87.4 | **89.6** |
| 70% | S-1 | **100.0** | 100.0 | 16.1 | 93.6 | **100.0** | 100.0 | **100.0** |
| | S-2 | 89.5 | 50.9 | 26.3 | 86.0 | **98.3** | 100.0 | 98.3 |
| | S-3 | 70.6 | 2.0 | 31.4 | 3.9 | **9.8** | **15.7** | 13.7 |
| | MK-1 | 79.2 | 33.3 | 35.4 | 75.0 | **87.5** | 97.9 | **97.9** |
| | MK-2 | 20.0 | 0.0 | 22.2 | 24.4 | **44.4** | 53.3 | **57.8** |
| | MK-3 | 17.0 | 8.5 | 19.2 | 8.5 | **23.4** | 0.0 | **4.3** |
| | MQ | **79.8** | 26.4 | 33.7 | 49.5 | 70.2 | **94.7** | **94.7** |
| | MV | 56.6 | 40.8 | 25.0 | 42.8 | **62.5** | 92.8 | 91.5 |
| | Average | **64.1** | 32.7 | 26.2 | 48.0 | 62.0 | 69.3 | **69.8** |

Table 17: Results of all KV compression baselines on needle-in-a-haystack subtasks with Mistral-7B at 50% and 70% compression ratios (▾: compression ratio).

| ▾ | SubTask | ChunkKV | Knorm | Streaming LLM | SnapKV | CurDKV | AdaSnapKV | AdaCurDKV |
|---|---|---|---|---|---|---|---|---|
| | | | | **Non-Adaptive Methods** | | | **Adaptive Methods** | |
| 50% | S-1 | 87.1 | **95.2** | 37.1 | 50.0 | 69.4 | 69.4 | **71.0** |
| | S-2 | 86.0 | 0.0 | 40.4 | 45.6 | **93.0** | 56.1 | **89.5** |
| | S-3 | 33.3 | 0.0 | **51.0** | 2.0 | 21.6 | 2.0 | **15.7** |
| | MK-1 | 77.1 | 0.0 | 50.0 | 20.8 | **79.2** | 43.8 | **77.1** |
| | MK-2 | 35.6 | 0.0 | **48.9** | 24.4 | 15.6 | **53.3** | 33.3 |
| | MK-3 | **25.5** | 0.0 | 14.9 | 12.8 | 2.1 | **38.3** | 10.6 |
| | MQ | **79.8** | 0.5 | 40.9 | 16.4 | 70.7 | 33.7 | **70.2** |
| | MV | 79.0 | 0.7 | 47.4 | 19.1 | **82.2** | 32.9 | **81.6** |
| | Average | **62.9** | 12.0 | 41.3 | 23.9 | 54.2 | 41.2 | **56.1** |
| 70% | S-1 | **88.7** | 80.7 | 14.5 | 45.2 | 19.4 | **69.4** | 12.9 |
| | S-2 | 56.1 | 0.0 | 26.3 | 24.6 | **68.4** | 40.4 | **52.6** |
| | S-3 | 11.8 | 0.0 | **31.4** | 2.0 | 0.0 | **2.0** | 0.0 |
| | MK-1 | **35.4** | 0.0 | 27.1 | 18.8 | **35.4** | 20.8 | **37.5** |
| | MK-2 | 13.3 | 0.0 | **17.8** | 4.4 | 0.0 | **24.4** | 2.2 |
| | MK-3 | **17.0** | 0.0 | 10.6 | 8.5 | 0.0 | **19.2** | 0.0 |
| | MQ | 48.6 | 0.0 | 26.0 | 12.0 | **64.4** | 13.9 | **26.9** |
| | MV | 44.7 | 0.0 | 25.0 | 15.1 | **63.8** | 17.8 | **43.4** |
| | Average | **39.5** | 10.1 | 22.3 | 16.3 | 31.4 | **26.0** | 21.9 |

Table 18: Ablation of CurDKV on LongBench datasets. We calculated leverage score on different attention modules (key, value and key-value both) with and without random Gaussian projections (▾: compression ratio).

| ▾ | Task | Key | Key+random | Key-value | Key-value+random | Value | Value+random |
|---|---|---|---|---|---|---|---|
| 30% | NrtvQA | 24.0 | 24.9 | 31.6 | 31.9 | 30.6 | 30.1 |
| | Qasper | 24.8 | 27.5 | 49.5 | 48.1 | 48.4 | 48.5 |
| | MF-en | 40.9 | 43.2 | 57.1 | 55.6 | 56.3 | 56.9 |
| | HotpotQA | 51.6 | 54.5 | 58.0 | 56.9 | 57.0 | 58.0 |
| | 2WikiMQA | 38.7 | 44.9 | 48.6 | 48.4 | 50.3 | 50.7 |
| | Musique | 23.4 | 28.3 | 31.0 | 34.2 | 33.2 | 32.7 |
| | Gov Report | 30.9 | 32.0 | 33.7 | 33.6 | 34.6 | 34.3 |
| | QMSum | 24.3 | 24.4 | 24.8 | 25.2 | 25.1 | 24.7 |
| | Multi News | 10.6 | 11.1 | 26.6 | 26.8 | 26.7 | 26.9 |
| | TREC | 33.0 | 25.0 | 65.5 | 54.5 | 66.5 | 62.5 |
| | TriviaQA | 91.0 | 90.9 | 92.2 | 92.1 | 92.4 | 91.7 |
| | SAMSum | 36.8 | 38.8 | 41.6 | 39.9 | 40.2 | 40.0 |
| | Pcount | 2.5 | 7.0 | 8.0 | 9.2 | 11.4 | 12.1 |
| | Pre | 81.5 | 92.5 | 99.0 | 98.0 | 100.0 | 99.0 |
| | Lcc | 52.9 | 53.5 | 51.1 | 51.6 | 50.7 | 51.4 |
| | RB-P | 51.7 | 50.4 | 50.1 | 50.0 | 49.9 | 49.4 |
| | Average | 38.7 | 40.6 | 48.0 | 47.2 | 48.3 | 48.0 |
| 90% | NrtvQA | 20.4 | 21.7 | 28.5 | 20.9 | 27.8 | 25.4 |
| | Qasper | 15.3 | 15.6 | 28.5 | 25.8 | 29.7 | 24.9 |
| | MF-en | 26.7 | 28.4 | 53.3 | 53.2 | 34.7 | 34.3 |
| | HotpotQA | 25.1 | 28.0 | 38.4 | 39.7 | 47.2 | 46.9 |
| | 2WikiMQA | 20.9 | 21.7 | 33.3 | 30.4 | 30.7 | 31.1 |
| | Musique | 9.5 | 10.8 | 20.0 | 18.7 | 19.7 | 21.9 |
| | Gov Report | 14.7 | 19.1 | 24.1 | 24.0 | 26.8 | 25.9 |
| | QMSum | 18.9 | 19.9 | 21.5 | 21.8 | 22.3 | 22.3 |
| | Multi News | 8.6 | 15.2 | 22.1 | 21.9 | 23.1 | 22.8 |
| | TREC | 1.5 | 6.5 | 7.0 | 15.3 | 22.5 | 16.0 |
| | TriviaQA | 87.0 | 88.4 | 89.9 | 90.6 | 90.7 | 91.7 |
| | SAMSum | 32.5 | 32.2 | 30.4 | 27.9 | 37.2 | 33.2 |
| | Pcount | 0.0 | 0.0 | 1.0 | 1.0 | 5.7 | 5.6 |
| | Pre | 3.5 | 6.0 | 25.0 | 26.0 | 61.0 | 49.0 |
| | Lcc | 33.5 | 32.0 | 39.7 | 38.2 | 34.9 | 35.3 |
| | RB-P | 55.3 | 55.1 | 54.0 | 53.6 | 49.6 | 51.9 |
| | Average | 23.3 | 25.0 | 32.3 | 31.8 | 35.2 | 33.7 |

This confirms that CUR-based token selection inherits the cache sparsity advantages of heuristic methods like SnapKV.

**Prefill Time.** SnapKV maintains flat prefill latency regardless of compression. For instance, at 128K tokens, prefill time remains consistently around 10 seconds across all compression levels (Figure 6b). CurDKV by contrast, incurs higher latency due to leverage score computation, rising from 10 seconds at 0% compression to about 14-15 seconds at 60–80% compression for 128K tokens

Table 19: Results on the needle-in-a-haystack benchmark (128K context length) at 30% and 50% compression with LLaMA-8B model.

| ▾ | Task | ChunkKV | KNorm | SnapKV | StreamingLLM | CurDKV |
|---|------|---------|-------|--------|--------------|--------|
| | S-1 | **100.00** | **100.00** | **100.00** | 55.56 | 88.89 |
| | S-2 | 87.50 | 87.50 | **100.00** | 62.50 | 87.50 |
| | S-3 | **90.91** | 81.82 | 63.64 | 54.55 | 81.82 |
| | MK-1 | 92.31 | 69.23 | **100.00** | 69.23 | 84.62 |
| 30% | MK-2 | 50.00 | 0.00 | 50.00 | 78.57 | **92.86** |
| | MK-3 | 20.00 | 10.00 | 10.00 | **20.00** | 10.00 |
| | MQ | 95.00 | 82.50 | **100.00** | 70.00 | **97.50** |
| | MV | 92.86 | 82.14 | 92.86 | 67.86 | **96.43** |
| | Average | 78.57 | 64.15 | 77.06 | 59.78 | **79.95** |
| | S-1 | **100.00** | **100.00** | **100.00** | 44.44 | **100.00** |
| | S-2 | 62.50 | 62.50 | **100.00** | 50.00 | 75.00 |
| | S-3 | **72.73** | 36.36 | 9.09 | 36.36 | 36.36 |
| | MK-1 | 84.62 | 23.08 | **92.31** | 53.85 | 84.62 |
| 50% | MK-2 | 35.71 | 0.00 | 42.86 | 50.00 | **57.14** |
| | MK-3 | 20.00 | 0.00 | 10.00 | 10.00 | 10.00 |
| | MQ | 85.00 | 45.00 | **95.00** | 40.00 | **90.00** |
| | MV | 82.14 | 46.43 | 92.86 | 39.29 | **96.43** |
| | Average | 67.84 | 39.17 | 67.77 | 40.49 | **68.69** |

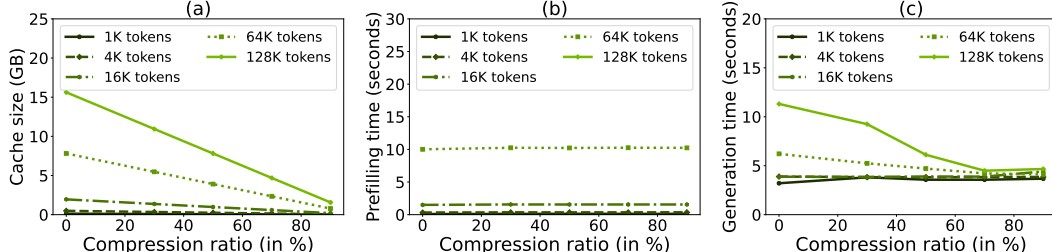

Figure 6: Prefilling and generation statistics of SnapKV for different sequence lengths with LLaMA-8B.

(Figure 4b). For shorter sequences like 4K or 16K, the difference is marginal, under 1 second for both methods.

**Generation Time.** `CurDKV` demonstrates a stronger reduction in generation time, particularly for long sequences. For 128K-token inputs, generation latency drops from ∼12 seconds at 0% compression to just ∼4.5 seconds at 80% compression (Figure 4c). SnapKV also sees a drop from ∼15 seconds to ∼6 seconds, but exhibits irregular behavior for mid-length sequences (e.g., 64K shows a flat profile between 40–80% compression in Figure 6c).

While SnapKV is faster to prefill, `CurDKV` yields larger reductions in generation time, especially for long sequences (e.g., 128K tokens) and high compression levels. Its additional prefill cost roughly 3–5 seconds at worst is offset by its smoother and more aggressive runtime scaling, making it a more efficient choice for long-context inference when accuracy and latency matter.

Table 20: Sensitivity and ablation analysis for `CurDKV` on selected LongBench tasks.

(a) **Sensitivity to safeguard parameter** $\alpha$

| ▼ | $\alpha$ | 2WikiMQA | HotpotQA | MF-en | Qasper | Avg |
|---|---|---|---|---|---|---|
| | 0.1 | 49.13 | 58.74 | 55.82 | 48.36 | 53.01 |
| 30% | 0.5 | 49.13 | 58.74 | 55.82 | 48.36 | 53.01 |
| | 0.9 | 48.99 | 58.67 | 56.33 | 48.46 | 53.11 |
| | 0.1 | 34.50 | 45.91 | 30.80 | 28.75 | 34.99 |
| 90% | 0.5 | 32.73 | 46.61 | 30.15 | 29.83 | 34.83 |
| | 0.9 | 32.27 | 46.44 | 31.37 | 28.99 | 34.77 |

(b) **Ablation on Gaussian projection dimension** $r$

| ▼ | $r$ | 2WikiMQA | HotpotQA | MF-en | Qasper | Avg |
|---|---|---|---|---|---|---|
| | 5 | 50.99 | 58.77 | 55.48 | 47.97 | 53.30 |
| 30% | 10 | 50.71 | 59.21 | 55.71 | 47.33 | 53.24 |
| | 50 | 49.31 | 57.85 | 54.78 | 47.83 | 52.44 |
| | 100 | 52.07 | 58.23 | 56.41 | 49.52 | 54.06 |
| | 5 | 28.45 | 41.80 | 29.29 | 25.50 | 31.26 |
| 90% | 10 | 35.04 | 41.98 | 29.94 | 26.02 | 33.24 |
| | 50 | 34.25 | 45.48 | 32.30 | 28.64 | 35.17 |
| | 100 | 35.30 | 46.52 | 33.15 | 28.13 | 35.77 |

