# OpenReview forum: "Value-Guided KV Compression for LLMs via Approximated CUR Decomposition"
_NeurIPS.cc/2025/Conference — NeurIPS 2025 poster_

### Official Review · Reviewer_hDao · 2025-06-29

**Clarity:** 3
**Significance:** 3
**Originality:** 3
**Rating:** 4
**Confidence:** 3

**Summary:**

This paper introduces CurDKV, a method which compresses KV-cache to reduce the inference-time memory comsumption for transformer-based models. CurDKV takes into account the importance of both K and V simultaneously and introduces a novel KV compression method based on CUR decomposition, in order to retain the most important KV-cache. In addition, CurDKV introduces Gaussian projections to compute approximate scores, further improving the efficiency of the KV-cache compresses process.

**Questions:**

* In Algorithm 1, CurDKV uses the product of K and V leverage scores, which assigns equal importance to K and V, which seems inconsistent with the earlier motivation that V is more significant. Can authors explain more about this point?
* In Figure 4(b) and fig 5(b), where is the data line for 128K tokens?

**Ethical Concerns:**

["NO or VERY MINOR ethics concerns only"]

**Final Justification:**

The authors solved most of my concerns in detail during the rebuttal period. Hence, I will keep my positive score as "borderline accept".

**Limitations:**

My questions are listed above and there is no other major limitations.

**Paper Formatting Concerns:**

There is no major formatting issues in this paper.

**Quality:**

3

**Strengths And Weaknesses:**

**Strengths**

* This paper leverages CUR decomposition to approximate the principal subspace of the attention output, which theoretically better preserves the LLM capacity. It is a good idea to use CUR decomposition to select KV cache based on the leverage score, considering both key and value.
* CurDKV is compatible with existing techniques such as FlashAttention and GQA.

**Weaknesses**

* In Section 3.1, the motivation for value-guided compression is not clear enough, as key can be equally important to the attention output. Moreover, the sentence "for higher compression levels, the second term becomes negligible" is a bit hard to follow.
* In Section 4.4, the paper analyze computational efficiency of CurDKV and lacks a comprehensive comparison with baseline methods. Since CurDKV introduces extra computation such as Gaussian projection and score combination, it is better to compare the end-to-end efficiency with other KV cache compression baselines.
* This paper uses Gaussian projections with $r = 20$, but lacks more analysis around Gaussian projections like how different r can impact the accuracy-efficiency trade-off. This ablation study will be very useful and this can be a minor weakness.

---

> ### Author Rebuttal · Authors · 2025-07-30
>
> We thank the reviewer for giving us a comprehensive analysis of our work and highlighting key weaknesses of our manuscript. We address some of the concerns below and hope that the discussion resolves all of them.
>
> **Clarification of the claims in Section 3.1 and a motivation for value-guided compression.**
>
> Here, we'll try to give a simpler motivation behind our approach of "value-guided compression" of KV caches. Recall that the standard attention output computes $\text{softmax}(QK^T/C)V$, where $C$ is a suitable constant involving the dimension of the vector spaces in question. Most of the current SOTA KV-cache compression methods (`SnapKV`, `H20`, `AdaKV`, to name a few) focus on estimating the importance of a cache entry by only using *attention weights*, which are computed using corresponding query and key vectors. However, as the attention equation suggests, value vectors may also contribute significantly to the final attention output. Lemma 3.1 of the paper bounds the true Frobenius distance between the attention outputs of the compressed and uncompressed KV caches; as we see, the upper bound depends on how well the value matrix is approximated. This intuition suggests that a good KV cache eviction policy should also incorporate the use of value vectors (and not just the attention weights). Having said this, we want to emphasize on the fact that this _does not_ undermine the importance of key vectors. Infact, we provide ablations to show that a combination of leverage scores of both the key and value vectors provides the best generation results (Figure 3 and Table 17 of our paper). Our ablation results also show that the next best choice is to only use the leverage scores of value vectors, which further highlight the importance of value vectors in designing key-value pair eviction policies.
>
> Regarding the sentence, "*for higher compression levels, the second term becomes negligible*": note that the matrix $V'$ is the compressed version of $V$, where dropped value vectors are simply zeroed out. At higher compression levels, say above 70%, most of the rows of $V'$ will just be zeros. Inevitably, the Frobenius norm of the resultant matrix will decrease, and will slowly become insignificant compared to the first term of the upper bound of Lemma 3.1.
>
> **End-to-end computational efficiency comparison with other baselines.** As requested by the reviewer, we run experiments to compare the end-to-end runtimes of `CurDKV` against other baselines.
>
> We compare four methods: exact `CurDKV` (where exact leverage scores are computed), random `CurDKV` (where the random Gaussian projection is used to project the key-value vectors, `SnapKV` and `ChunkKV`. We do comparisons on a context length of 128K tokens and generated length of 100 tokens, and we use compression ratios of $30$% and $90$%. We report the total runtime, prefill time and the generation time for all methods. Here are the detailed results:
>
> **Table 1:** Runtime results for a context length of 128K tokens and a compression ratio of $30$%.
>
> | Method          | Length | Total time  (in sec)     | Prefill time  (in sec)   | Generation time (in sec) |
> | --------------- | ------ | ---------------- | ---------------- | ---------------- |
> | Exact `CurDKV`  | 128K | 55.39 | 46.51 | 8.88 |
> | Random `CurDKV` | 128K | **44.76**  | 35.17 | 9.59 |
> | `SnapKV`        | 128K | 44.97  | **33.37** | 11.59 |
> | `ChunkKV`       | 128K | 54.36 | 50.53 | **3.83** |
>
> As we see from the above table, `CurDKV` with a random Gaussian projection outperforms the baselines for a compression ratio of $30$%. This is also intuitive, since the Gaussian projection reduces the effective dimension of the key-value pairs. This, combined with the computationally efficient leverage scores, leads to the overall smallest total time (prefill + generation) for random `CurDKV`.
>
> **Table 2:** Runtime results for a context length of 128K tokens and a compression ratio of $90$%.
> | Method          | Length | Total time       | Prefill time     | Generation time  |
> | --------------- | ------ | ---------------- | ---------------- | ---------------- |
> | Exact `CurDKV`  | 128K | 51.36 | 46.90 | **4.46** |
> | Random `CurDKV` | 128K | 42.82 | 35.41 | 7.41 |
> | `SnapKV`        | 128K | 44.50 | **33.59** | 10.91 |
> | `ChunkKV`       | 128K | **42.74** | 35.83 | 6.91 |
>
> For a high compression ratio of $90$%, random `CurDKV` still performs approximately the same as the best baseline (`ChunkKV` for this case). This, combined with the performance gains of `CurDKV`, makes it a competitive choice.
>
> **An ablation study on the dimension of the Gaussian projection.** As rightfully requested by the reviewer, we run an ablation study to determine the effect of the output dimension $r$ of the Gaussian projection matrix. Intuitively, we expect an overall increase in the generation quality as the dimension $r$ increases; this is due to the simple fact that as the dimension increases, the leverage scores of the projected vectors will better approximate the leverage scores of the original vectors. However, we still need to empirically verify this claim, and we do so below. We carry out these ablation studies for compression ratios of $30$% and $90$%. We run evaluations on the tasks in the **LongBench** benchmarks.
>
> **Table 3:** Ablation study on the output dimension $r$ of the Gaussian projection matrix for a compression ratio of $30\%$.
>
> | $r$ | LongBench/2wikimqa | LongBench/hotpotqa | LongBench/multifieldqa_en | LongBench/qasper | Average |
> | -------- | -------- |  -------- |  -------- |  -------- |  -------- |
> | 5     | 50.99     | 58.77     | 55.48     | 47.97     | 53.30     |
> | 10     | 50.71     | 59.21     | 55.71     | 47.33     | 53.24     |
> | 50     | 49.31     | 57.85     | 54.78     | 47.83     | 52.44    |
> | 100     | 52.07     | 58.23     | 56.41     | 49.52     | 54.06     |
>
> **Table 4:** Ablation study on the output dimension $r$ of the Gaussian projection matrix for a compression ratio of $90$%.
>
> | $r$ | LongBench/2wikimqa | LongBench/hotpotqa | LongBench/multifieldqa_en | LongBench/qasper | Average |
> | -------- | -------- |  -------- |  -------- |  -------- |  -------- |
> | 5     | 28.45     | 41.8     | 29.29     | 25.5     | 31.26   |
> | 10     | 35.04     | 41.98     | 29.94     | 26.02     | 33.24     |
> | 50     | 34.25     | 45.48     | 32.3     | 28.64     | 35.17     |
> | 100     | 35.3     | 46.52     | 33.15     | 28.13     | 35.77     |
>
> For both compression ratios, we observe that increasing $r$ improves the performance.  In Figure 3 in Section 4.2, we have discussed that CurDKV without random projection (Exact CUR) usually performs better than CurDKV with random projection. Having a large $r$ in the approximated leverage score computation approximates the exact leverage score computation. Therefore, the performance increases to that of the exact CUR. Moreover, this behavior is more prominent for larger compression ratios.
>
> **128K data line for Figures 4(b) and 5(b).**
>
> We thank the reviewer for pointing this out. Due to a limit in the y-axis (the maximum y value for prefill time is taken as 30 sec in the plots, but for 128K, it exceeds 40 sec), the plots for 128K were omitted from both plots. We will correct this in the revised version of the paper.

---

> > ### Comment · Reviewer_hDao · 2025-08-01
> > **Reply on the Authors' Rebuttal**
> >
> > Thanks for your rebuttal and results. I don't have further questions, and I'll keep my score.

---

### Official Review · Reviewer_NWDK · 2025-07-04

**Clarity:** 3
**Significance:** 3
**Originality:** 3
**Rating:** 4
**Confidence:** 3

**Summary:**

The authors propose CurDKV, a value-centric KV compression method that selects keys and values based on leverage scores computed from CUR matrix decomposition. CurDKV achieves up to 9.6% higher accuracy than state-of-the-art methods like SnapKV and ChunkKV under aggressive compression budgets on LLaMA and Mistral, while maintaining compatibility with FlashAttention and Grouped Query Attention. In addition to improved accuracy, CurDKV reduces generation latency by up to 40% at high compression, offering a practical speed-accuracy tradeoff.

**Questions:**

How does the method process the KV of generated tokens?
Does the method support vllm?

**Ethical Concerns:**

["NO or VERY MINOR ethics concerns only"]

**Limitations:**

Yes

**Quality:**

3

**Strengths And Weaknesses:**

Strengths:
1. The experiments are solid. The proposed method is better than strong baselines.
2. CurDKV is simple and effective without further training.
3. The paper is well-written and easy to read.

Weaknesses:
1. The proposed method is only tested on the tasks with longer context. It is not clear how the method works on the tasks that need to generate longer sequence, such as story writing, code generation etc. How does the method process the KV of generated tokens?
2. The paper claims the value guided method, while both key and value are used to compute scores in Algorithm 1. Need clarification on it. And it would be better to have an ablation study on which is important to guide topk indices.

---

> ### Author Rebuttal · Authors · 2025-07-30
>
> We thank the reviewer for raising important questions about our claim to develop "value-guided" KV compression methods. We are glad to know that the reviewer found our work strong, easy to understand, and practically effective. We address the concerns raised by the reviewer below and hope that this discussion resolves their queries.
>
> **How does CurDKV work in inference-time for longer generation tasks**
>
> Below, we detail how our method processes the Key-Value (KV) cache for tokens generated during inference -
>
> 1. At each generation step, the model generates a token and computes the corresponding query, key, and value vectors.
> 2. The newly generated token's key and value vectors are appended to the KV cache of the prefilled (context) tokens.
> 3. After appending the new token, the method periodically recomputes leverage scores for the cached keys and values (optionally, using the approximation via random Gaussian projections to ensure computational efficiency)
> 4. Tokens with the lowest combined leverage scores are evicted from the cache to maintain the predefined memory budget. The most informative tokens, identified via high leverage scores, are retained.
> 5. CurDKV explicitly preserves the first few tokens (attention sinks) as they typically carry critical global context relevant for subsequent generation steps.
>
> We further compare four KV compression methods: exact `CurDKV` (where exact leverage scores are computed), random `CurDKV` (where the random Gaussian projection is used to project the key-value vectors, `SnapKV` and `ChunkKV`. We do comparisons on a context length of 128K tokens and generated length of 100 tokens, and we use compression ratios of $30$% and $90$%. We report the total runtime, prefill time and the generation time for all methods. Here are the detailed results:
>
> **Table 1:** Runtime results for a context length of 128K tokens and a compression ratio of $30$%.
>
> | Method          | Length | Total time  (in sec)     | Prefill time  (in sec)   | Generation time (in sec) |
> | --------------- | ------ | ---------------- | ---------------- | ---------------- |
> | Exact `CurDKV`  | 128K | 55.39 | 46.51 | 8.88 |
> | Random `CurDKV` | 128K | **44.76**  | 35.17 | 9.59 |
> | `SnapKV`        | 128K | 44.97  | **33.37** | 11.59 |
> | `ChunkKV`       | 128K | 54.36 | 50.53 | **3.83** |
>
> As we see from the above table, `CurDKV` with a random Gaussian projection outperforms the baselines for a compression ratio of $30$%. This is also intuitive, since the Gaussian projection reduces the effective dimension of the key-value pairs. This, combined with the computationally efficient leverage scores, leads to the overall smallest total time (prefill + generation) for random `CurDKV`.
>
> **Table 2:** Runtime results for a context length of 128K tokens and a compression ratio of $90$%.
> | Method          | Length | Total time (in sec)      | Prefill time (in sec)    | Generation time (in sec) |
> | --------------- | ------ | ---------------- | ---------------- | ---------------- |
> | Exact `CurDKV`  | 128K | 51.36 | 46.90 | **4.46** |
> | Random `CurDKV` | 128K | 42.82 | 35.41 | 7.41 |
> | `SnapKV`        | 128K | 44.50 | **33.59** | 10.91 |
> | `ChunkKV`       | 128K | **42.74** | 35.83 | 6.91 |
>
> For a high compression ratio of $90$%, random `CurDKV` still performs approximately the same as the best baseline (`ChunkKV` for this case). This, combined with the performance gains of `CurDKV`, makes it a competitive choice.
>
> When we increase the generation length to 4K, the generation time increases for all compression methods due to periodic eviction/retention of key-value pairs.
>
> **Table 3:** Generation time results for a context length of 128K and generated 4K tokens.
> | Method          |  Compression ratio  | Generation time (in sec)  |
> | --------------- | ------ | ---------------- |
> | Exact `CurDKV`  | 30% | **464.9** |
> | Random `CurDKV` | 30% | 516.9 |
> | `SnapKV`        | 30% | 553.2 |
> | `ChunkKV`       | 30% | 539.3 |
> | Exact `CurDKV`  | 90% | 656.8 |
> | Random `CurDKV` | 90% | **536.9** |
> | `SnapKV`        | 90% | 553.4 |
> | `ChunkKV`       | 90% | 545.6 |
>
> At a 30\% compression ratio, exact CurDKV exhibits the fastest generation, whereas, for a 90\% compression ratio, random CurDKV achieves the fastest generation. At moderate compression ratios, the Exact CurDKV precisely identifies and retains highly informative keys and values through exact CUR decomposition; therefore, reducing the need for repeatedly decomposing the key-value pairs on generated tokens. At a higher compression rate, more repeated computation of key-value eviction is needed; therefore, under those circumstances, the effectiveness of low-rank projections increases, leading to lower generation time.
>
> **On the usage of both the key and value matrices, and the claim of the paper.**
> We appreciate the reviewer for seeking clarification for the main claim of the paper, as we believe it is the main contribution of this work. Most of the existing SOTA literature on KV cache compression methods (`SnapKV`, `H20`, `AdaKV`, to name a few) focus on estimating the importance of a cache entry by using _attention weights_, which rely only on the corresponding query and key vectors. However, as we show in Lemma 3.1 of our paper, the true Frobenius distance between the exact attention output and the compressed attention output also depends on how well the compressed value matrix approximates the original value matrix; intuitively, this happens because the query and key vectors are absorbed by the softmax operator. This motivates the design of eviction policies which also incorporate value vectors. At the same time, we point out that this analysis _does not undermine_ the importance of the corresponding key vectors. Infact, we provide ablation studies to empirically show that the combined scores of both the key and value produces the best results (Figure 3 and Table 17 of our paper). As our ablations also show, the next best choice is to use the leverage scores of only the value vectors. Overall, through our paper, we aim to establish the importance of incorporating information gained from value vectors in developing key-value eviction policies.
>
> **`vllm` support**
>
> We thank the reviewer for raising this important point regarding practical compatibility and integration. CurDKV is fully supported by NVIDIA’s KVPress library, which is specifically designed to implement and benchmark KV cache pruning strategies within Hugging Face Transformers-based pipelines. However, at present, KVPress (and thus CurDKV) does not natively support vLLM. The vLLM inference backend manages KV caches differently and currently has its own separate mechanisms (e.g., KV-Compress, LeanKV) for cache compression. While direct compatibility between CurDKV and vLLM would require additional engineering effort, methodologically, CurDKV can be adapted to work with vLLM by modifying vLLM’s cache eviction and retention mechanisms accordingly.

---

> > ### Author Response · Authors · 2025-08-06
> > **Please check our responses to your comments**
> >
> > Dear Reviewer NWDK,
> >
> > Thank you for your valuable and insightful comments. We have carefully addressed each of your points and made every effort to incorporate your suggestions into the revised version of our paper. We are confident that our responses have resolved the concerns you raised.
> >
> > We would greatly appreciate it if you could kindly review our responses and consider reassessing our submission. As the discussion phase is concluding soon, we hope to engage in a productive dialogue with you before it ends.
> >
> > Thank you once again for your time and consideration.
> >
> > Thanks
> >
> > Authors

---

> > > ### Author Response · Authors · 2025-08-08
> > > **Requesting Reviewer for Checking Our Responses**
> > >
> > > Dear Reviewer,
> > >
> > > As only 24 hours are left till the discussion period ends, we once again request you to kindly check our responses. We have carefully addressed each of your points and made every effort to incorporate your suggestions into the revised version of our paper. We are confident that our responses have resolved the concerns you raised.
> > >
> > > We would greatly appreciate it if you could kindly review our responses and consider reassessing our submission.

---

> > > > ### Comment · Area_Chair_oJAd · 2025-08-08
> > > >
> > > > Dear Reviewer NWDK,
> > > >
> > > > The discussion period is now within 24 hours. Please check out the authors' response ASAP!
> > > >
> > > > Best,
> > > >
> > > > AC

---

### Official Review · Reviewer_1huD · 2025-07-05

**Clarity:** 2
**Significance:** 2
**Originality:** 3
**Rating:** 4
**Confidence:** 3

**Summary:**

The paper propose CurDKV for compressing the key-value (KV) cache, a value-centric strategy. They estimate each token’s leverage score via a (fast, projected) CUR matrix decomposition of the attention output and keep only the most influential keys and values.

**Questions:**

see weaknesses

**Ethical Concerns:**

["NO or VERY MINOR ethics concerns only"]

**Final Justification:**

After reading the authors’ rebuttal and the comments from other reviewers, most of my concerns, including some experimental justifications involving larger models, have been addressed.

 Therefore, I am raising my score by one point.

**Limitations:**

Yes, the paper discusses the pertinent limitations and outlines directions for future work.

**Quality:**

2

**Strengths And Weaknesses:**

Strengths
1. CUR-based leverage scores + Gaussian projection yield an O(n d r) lightweight scoring pass; algorithm integrates with FlashAttention & Grouped-Query Attention without kernel changes.

2. Memory scales predictably; although pre-fill cost rises by ~3–5 s for 128 K contexts, generation is faster for long sequences


Weaknesses

1. Static Compression Only. CurDKV freezes the cache after prefill; it cannot react if later tokens shift relevance, and it adds noticeable prefill overhead for very long inputs

2. Limited Model Coverage. It is better to provide more results on bigger models. The results use 7 B / 8 B LLaMA/Mistral; scalability to 34 B–70 B or MoE models is untested.

3. Hyper-Parameter Sensitivity. Adaptive safeguard α (0.20) is fixed; sensitivity analysis would help.

---

> ### Author Rebuttal · Authors · 2025-07-30
>
> We thank the reviewer for giving us valuable insights to the potential weaknesses of our work, and we aim to address some of those concerns below. We hope the following discussion clarifies some concerns raised by the reviewer.
>
> **`CurDKV` as a static compression method.** It is true that `CurDKV` is a static compression method, in the sense that once a key-value pair of the context is evicted from the cache, `CurDKV` never recovers it (even if future tokens shift relevance). However, we'd like to point out that some of the best-performing baselines which we compare against are also static in this sense: `H2o`, `SnapKV`, `StreamingLLM` and `AdaKV` to name a few. Infact, any reasonable KV cache compression technique must be static in this sense, because the recovery of an evicted key-value pair from the cache must inevitably require either 1) storing the key-value pair in GPU memory or 2) recomputing the key-value pair, both of which may have significant overhead for larger context lengths.
>
> **Results on bigger models.** Per request of the reviewer, we tested `CurDKV` on larger models; particularly, we chose the Qwen3 14B and Qwen2.5 32B models, and tested the efficacy of `CurDKV` on these models against other strong baselines. As in the paper, we report results on tasks from the **LongBench** benchmark. We report results for compression ratios of $30$% and $90$% respectively.
>
> **Table 1:** **LongBench** evaluation results for the Qwen 14B model for a compression ratio of $30$%.
>
> | Method | LongBench/2wikimqa | LongBench/hotpotqa | LongBench/multifieldqa_en | LongBench/qasper | Average |
> | -------- | -------- | -------- | -------- | -------- | -------- |
> | `ChunkKV`     | **28.71**     | 33.53     | 30.41     | 24.84     | 29.37     |
> | `KNorm`     | 26.78     | 31.44     | 29.27     | 21.89     | 27.34     |
> | `SnapKV`     | 28.45     | **34.56**     | **32.11**     | 25.38     | **30.12**     |
> | `StreamingLLM`     | 20.56     | 26.67     | 22.33     | 22.08     | 22.91     |
> | `CurDKV`     | 27.97     | 31.58     | 31.95     | **27.04**     | 29.63     |
>
> As we see from the above table, `SnapKV` seems to perform the best on average for the **LongBench** tasks. However, `CurDKV` is not far, and consistently performs close to the best baseline.
>
> **Table 2:** **LongBench** evaluation results for the Qwen 14B model for a compression ratio of $90$%.
>
> | Method | LongBench/2wikimqa | LongBench/hotpotqa | LongBench/multifieldqa_en | LongBench/qasper | Average |
> | -------- | -------- | -------- | -------- | -------- | -------- |
> | `ChunkKV`     | 12.39     | 21.09     | 17.03     | 9.25     | 14.94     |
> | `KNorm`      | 10.04     | 9.73     | 13.06     | 7.2     | 10.00     |
> | `SnapKV`       | 13.37     | 21.34     | 18.25     | 9.56     | 15.63     |
> | `StreamingLLM`     | 9.56     | 14.51     | 14.66     | 10.22     | 12.24     |
> | `CurDKV`     | **20.88**     | **21.54**     | **22.08**     | **16.23**     | **20.18**     |
>
> The effect of `CurDKV` seems to be more prominent for the regime of high compression ratios. For instance, in the above table ($90$% compression), `CurDKV` significantly outperforms other baselines by a significant margin. Below, we see a more prominent effect for an even larger model (a 34B quantized model).
>
> **Table 3:** **LongBench** evaluation results for the Qwen 32B model for a compression ratio of $30$%.
>
> | Method | LongBench/2wikimqa | LongBench/hotpotqa | LongBench/multifieldqa_en | LongBench/qasper | Average |
> | -------- | -------- | -------- | -------- | -------- | -------- |
> | `ChunkKV`     | 51.64     | 58.64     | 45.41     | 42.46     | 49.54     |
> | `KNorm`     | 40.14     | 46.94     | 43.39     | 37.73     | 42.05     |
> | `SnapKV`     | 53.85     | 57.52     | 48.56     | 41.86     | 50.45     |
> | `StreamingLLM`     | 34.1     | 47.15     | 32.61     | 43.11     | 39.24     |
> | `CurDKV`     | **58.07**     | **62.6**     | **50.42**     | **47.75**     | **54.71**     |
>
> Here again, we see that `CurDKV` consistently outperforms other baselines by a significant margin.
>
> **Table 4:** **LongBench** evaluation results for the Qwen 32B model for a compression ratio of $90$%.
>
> | Method | LongBench/2wikimqa | LongBench/hotpotqa | LongBench/multifieldqa_en | LongBench/qasper | Average |
> | -------- | -------- | -------- | -------- | -------- | -------- |
> | `ChunkKV`     | 24.24     | 37.85     | 24.31     | 13.62     | 25.00     |
> | `KNorm`     | 6.14     | 3.8     | 13.12     | 7.02     | 7.52     |
> | `SnapKV`     | 22.8     | 36.29     | 23.04     | 14.29     | 24.10     |
> | `StreamingLLM`     | 18.54     | 30.38     | 22.03     | 14.56     | 21.38     |
> | `CurDKV`     | **40.59**     | **43.51**     | **31.24**     | **16.1**     | **32.86**     |
>
> And yet again, we see that for the 32B model quantized model, the gains for `CurDKV` in a high compression ratio regime are even more prominent. This raises a further interesting question: *do value-guided KV caching methods perform better for quantized models?* We leave this question for a future work.
>
> **On the sensitivity of the safeguard parameter ($\alpha$).** As requested by the reviewer, here we do experiments to run a sensitivity analysis on the safeguard parameter $\alpha$. To recall, for KV compression methods that distribute the per-head budget adaptively (aka the `AdaKV` technique [1]), the parameter $\alpha$ controls the proportion of KV pairs to preserve in each head (see line 28 of the `KVPress` implementation at https://github.com/NVIDIA/kvpress/blob/main/kvpress/presses/adakv_press.py). The default value of $\alpha$ as proposed by the authors of `AdaKV` is $0.20$, which we also use in our implementation of `AdaCurDKV`. Tables 5 and 6 below contain the results of our ablations.
>
>
> **Table 5:** Sensitivity analysis of the safeguard parameter $\alpha$ for **LongBench** tasks at a compression ratio of $30$%. We compare the two adaptive techniques: `AdaSnapKV` and `AdaCurDKV`.
>
> | $\alpha$ | Method | LongBench/2wikimqa | LongBench/hotpotqa | LongBench/multifieldqa_en | LongBench/qasper | Average |
> | -------- | -------- | -------- |  -------- |  -------- |  -------- |  -------- |
> | 0.1     | `AdaCurDKV`     | 49.13     | 58.74     | 55.82     | 48.36     | 53.01     |
> | 0.1     | `AdaSnapKV`     | 50.58     | 59.93     | 55.28     | 45.74     | 52.88     |
> | 0.5     | `AdaCurDKV`     | 49.13     | 58.74     | 55.82     | 48.36     | 53.01     |
> | 0.5     | `AdaSnapKV`     | 51.34     | 59.55     | 55.61     | 45.84     | 53.08     |
> | 0.9     | `AdaCurDKV`     | 48.99     | 58.67     | 56.33     | 48.46     | 53.11     |
> | 0.9     | `AdaSnapKV`     | 51.06     | 57.9     | 54     | 45.72     | 52.17     |
>
> **Table 6:** Sensitivity analysis of the safeguard parameter $\alpha$ for **LongBench** tasks at a compression ratio of $90$%. We compare the two adaptive techniques: `AdaSnapKV` and `AdaCurDKV`.
>
> | $\alpha$ | Method | LongBench/2wikimqa | LongBench/hotpotqa | LongBench/multifieldqa_en | LongBench/qasper | Average |
> | -------- | -------- | -------- |  -------- |  -------- |  -------- |  -------- |
> | 0.1     | `AdaCurDKV`     | 34.5     | 45.91     | 30.8     | 28.75     | 34.99     |
> | 0.1     | `AdaSnapKV`     | 26.7     | 46.18     | 26.86     | 21.53     | 30.32    |
> | 0.5     | `AdaCurDKV`     | 32.73     | 46.61     | 30.15     | 29.83     | 34.83     |
> | 0.5     | `AdaSnapKV`     | 27.15     | 46.07     | 26.18     | 20.39     | 29.95     |
> | 0.9     | `AdaCurDKV`     | 32.27     | 46.44     | 31.37     | 28.99     | 34.77     |
> | 0.9     | `AdaSnapKV`     | 23.48     | 45.27     | 26.46     | 21.65     | 29.21     |
>
> For both 30% and 90% compression ratios, we observe only a marginal performance difference for different values of $\alpha$, indicating that the adaptive KV compression methods (including `AdaCurDKV`) are very robust to the selection of this parameter.
>
> **References**
>
> [1] Ada-KV: Optimizing KV Cache Eviction by Adaptive Budget Allocation for Efficient LLM Inference, Feng et al., 2025.

---

> > ### Author Response · Authors · 2025-08-06
> > **Please check our responses to your comments**
> >
> > Dear Reviewer 1huD,
> >
> > Thank you for your valuable and insightful comments. We have carefully addressed each of your points and made every effort to incorporate your suggestions into the revised version of our paper. We are confident that our responses have resolved the concerns you raised.
> >
> > We would greatly appreciate it if you could kindly review our responses and consider reassessing our submission. As the discussion phase is concluding soon, we hope to engage in a productive dialogue with you before it ends.
> >
> > Thank you once again for your time and consideration.
> >
> > Thanks
> >
> > Authors

---

> > > ### Author Response · Authors · 2025-08-08
> > > **Requesting Reviewer for Checking Our Responses**
> > >
> > > Dear Reviewer,
> > >
> > > As only 24 hours are left till the discussion period ends, we once again request you to kindly check our responses. We have carefully addressed each of your points and made every effort to incorporate your suggestions into the revised version of our paper. We are confident that our responses have resolved the concerns you raised.
> > >
> > > We would greatly appreciate it if you could kindly review our responses and consider reassessing our submission.

---

> > > > ### Comment · Area_Chair_oJAd · 2025-08-08
> > > >
> > > > Dear Reviewer 1huD,
> > > >
> > > > The discussion period is now within 24 hours. Please check out the authors' response ASAP!
> > > >
> > > > Best, AC

---

> > > > > ### Comment · Reviewer_1huD · 2025-08-08
> > > > >
> > > > > After reading the rebuttal and other review information, most of my concerns have been addressed, and I will maintain my positive score.

---

### Official Review · Reviewer_kZ3Z · 2025-07-07

**Clarity:** 3
**Significance:** 3
**Originality:** 3
**Rating:** 4
**Confidence:** 4

**Summary:**

This paper introduce CurdKV, that rethinks kv cache compression by consider not only keys but also values via a leverage score metric. Specifically, Curdkv uses cur decomposition to identify more important individual keys and values via leverage score, where a importance is measures for rows and cols of the key and value matrices. In practice, this process is done on a group level to compatible with GQA. And to avoid svd it use a random project for the decomposition to reduce overhead. At last, an adaptive budget version of curdkv is proposed.

**Questions:**

No

**Ethical Concerns:**

["NO or VERY MINOR ethics concerns only"]

**Final Justification:**

I'll keep my score.

**Quality:**

3

**Strengths And Weaknesses:**

Strengths

1. The proposed kv cache include values in the kv cache compression process, that were overlooked previously. And this is well motivated by fig 1 and introduction.

2. The proposed method to use cur decomposition is solid when identifying the important rows and cols.

3. To use a random projection for leverage score computing seems a good trick to avoid svd. And the extra cost involved is acceptable. Plus this importance evaluation process can be fit inside flash-attn, which is important in real applications.

4. Solid overall performance on longbench and the NIAH test.

Weakness

1. Algo 1 is not very clear to me. How's the leverage score is used for the decomposition?

2. The max length in the experiments is rather limited. I would expect to test upto 128k on RULER.

3. Why not include a latency comparision with and without curd kv? to show the exact overhead used in computing leverage score and the ranking process.

---

> ### Author Rebuttal · Authors · 2025-07-30
>
> We thank the reviewer for giving us a comprehensive analysis of our work. We are grateful that the reviewer found our approach of using values to compress KV caches well-motivated. Below, we would like to address all concerns of the reviewer and hope that they answer all their questions.
>
> **A clearer description of Algorithm 1.** Here we give a clearer description of the algorithm, with a particular focus on how leverage scores are used to compute the decomposition.
>
> 1. Consider a set of key and value matrices $K\in\mathbb{R}^{k\times d}$ and $V\in\mathbb{R}^{k\times d}$, i.e $k$ key-value pairs, each of dimension $d$.
> 2. Project key-value vectors do a vector of dimension $r$ (we use $r = 20$) via a Gaussian projection matrix $G\in\mathbb{R}^{d\times r}$ (lines 2-3 of the algorithm). The significance of this step is to reduce the overall run-time complexity (computing exact leverage scores requires SVD).
> 3. After this, we compute *leverage scores* using the new key and value vectors (line 4 of the algorithm). In simple words, these scores are just row norms of the key-value matrices.
> 4. The key-value leverage scores are then combined (line 5) and normalized (line 6).
> 5. Finally, the indices with the top-$(k - s)$ scores are computed, and the keys and values for these indices are retained in the KV cache (lines 8-10 of the algorithm).
> 6. Note that we preserve the first $s$ tokens in the context due to the prevalence of *attention sinks*. [1]
>
> ***Note***: We've further provided an ablation study involving different values of $r$ for reviewer hDao. TL;DR: we observe only marginal differences when using different values of $r$, and the Gaussian approximation improves as $r$ increases (which is intuitive).
>
> **Regarding the maximum length in experiments for the RULER benchmark.** As per the reviewer's request, we ran NIAH experiments on the **RULER** benchmark with a context length of 128K tokens. We've reported the results in the table given below. As in the main paper, we compare `CurDKV` with the `ChunkKV`, `KNorm`, `SnapKV` and `StreamingLLM` baselines. We carry out the experiments with compression ratios of $30$%.
>
> **Table 1:** Results on the **RULER** benchmark with 128K context length with a compression ratio of $30$%.
>
> Task/Method| `ChunkKV` | `KNorm` | `SnapKV` | `StreamingLLM` | `CurDKV` |
> | -------- | -------- | -------- | -------- | -------- |  -------- |
> | `Accuracy.niah_multikey_1.string_match` | 92.31     | 69.23     | **100**     |  69.23 |  84.62 |
> | `Accuracy.niah_multikey_2.string_match` | 50     | 0     | 50     |  78.57 |  **92.86** |
> | `Accuracy.niah_multikey_3.string_match` | **20**     | 10     | 10     |  **20** |  10 |
> | `Accuracy.niah_multiquery.string_match` | 95     | 82.5     | **100**     |  70 |  97.5 |
> | `Accuracy.niah_multivalue.string_match` | 92.86     | 82.14     | 92.86     |  67.86 |  **96.43** |
> | `Accuracy.niah_single_1.string_match` | **100**     | **100**     | **100**     |  55.56 |  88.89 |
> | `Accuracy.niah_single_2.string_match` | 87.5     | 87.5     | **100**     |  62.5 |  87.5 |
> | `Accuracy.niah_single_3.string_match` | **90.91**     | 81.82     | 63.64     |  54.55 |  81.82 |
> | `Average` | 78.572    | 64.15     | 77.06    |  59.78 |  **79.95** |
>
> **Table 2:** Results on the **RULER** benchmark with 128K context length with a compression ratio of $50$%.
>
> Task/Method| `ChunkKV` | `KNorm` | `SnapKV` | `StreamingLLM` | `CurDKV` |
> | -------- | -------- | -------- | -------- | -------- |  -------- |
> | `Accuracy.niah_multikey_1.string_match`       | 84.62  | 23.08 | **92.31**  | 53.85        | 84.62 |
> | `Accuracy.niah_multikey_2.string_match`       | 35.71  | 0     | 42.86  | 50.00        | **57.14** |
> | `Accuracy.niah_multikey_3.string_match`       | **20.00**  | 0     | 10.00  | 10.00        | 10.00 |
> | `Accuracy.niah_multiquery.string_match`       | 85.00  | 45.00 | **95.00**  | 40.00        | 90.00 |
> | `Accuracy.niah_multivalue.string_match`       | 82.14  | 46.43 | 92.86  | 39.29        | **96.43** |
> | `Accuracy.niah_single_1.string_match`         | **100.00** | **100.0** | **100.00** | 44.44        | **100.00** |
> | `Accuracy.niah_single_2.string_match`         | 62.50  | 62.50 | **100.00** | 50.00        | 75.00 |
> | `Accuracy.niah_single_3.string_match`         | **72.73**  | 36.36 | 9.09   | 36.36        | 36.36 |
> | `Average`                                 | 67.84  | 39.17 | 67.77  | 40.49        | **68.69** |
>
> As we see from the above results, `CurDKV` consistently either outperforms other methods or performs in a way sufficiently comparable to the best baseline for a given task, leading to the overall best average score.
>
> **Note:** For NIAH 128K experiments, we chose 30% and 50% compression ratios for comparing the KV compression methods.  At higher compression ratios (>50%), most of the methods' performance drops poorly (almost to 0 for most tasks) due to a lack of the appropriate context length. This behavior is more prevalent in the 128K context than in shorter context NIAH tasks (reported in the paper).
>
> **A latency comparison with and without `CurDKV`.** Per request of the reviewer, here we provide a latency comparison of other baselines against `CurDKV`. We compare four methods: exact `CurDKV` (where exact leverage scores are computed), random `CurDKV` (where the random Gaussian projection is used to project the key-value vectors, `SnapKV` and `ChunkKV`. We do comparisons on a context length of 128K tokens and generated length of 100 tokens, and we use compression ratios of $30$% and $90$%. We report the total runtime, prefill time, and the generation time for all methods. Here are the detailed results:
>
> **Table 3:** Runtime results for a context length of 128K tokens and a compression ratio of $30$%.
>
> | Method          | Length | Total time  (in sec)     | Prefill time  (in sec)   | Generation time (in sec) |
> | --------------- | ------ | ---------------- | ---------------- | ---------------- |
> | Exact `CurDKV`  | 128K | 55.39 | 46.51 | 8.88 |
> | Random `CurDKV` | 128K | **44.76**  | 35.17 | 9.59 |
> | `SnapKV`        | 128K | 44.97  | **33.37** | 11.59 |
> | `ChunkKV`       | 128K | 54.36 | 50.53 | **3.83** |
>
> As we see from the above table, `CurDKV` with a random Gaussian projection outperforms the baselines for a compression ratio of $30$%. This is also intuitive, since the Gaussian projection reduces the effective dimension of the key-value pairs. This, combined with the computationally efficient leverage scores, leads to the overall smallest total time (prefill + generation) for random `CurDKV`.
>
> **Table 4:** Runtime results for a context length of 128K tokens and a compression ratio of $90$%.
> | Method          | Length | Total time       | Prefill time     | Generation time  |
> | --------------- | ------ | ---------------- | ---------------- | ---------------- |
> | Exact `CurDKV`  | 128K | 51.36 | 46.90 | **4.46** |
> | Random `CurDKV` | 128K | 42.82 | 35.41 | 7.41 |
> | `SnapKV`        | 128K | 44.50 | **33.59** | 10.91 |
> | `ChunkKV`       | 128K | **42.74** | 35.83 | 6.91 |
>
> For a high compression ratio of $90$%, random `CurDKV` still performs approximately the same as the best baseline (`ChunkKV` for this case). This, combined with the performance gains of `CurDKV`, makes it a competitive choice.
>
> **References**
>
> [1] Efficient Streaming Language Models with Attention Sinks, Xiao et al., 2024.

---

> > ### Author Response · Authors · 2025-08-06
> > **Please check our responses to your comments**
> >
> > Dear Reviewer kZ3Z,
> >
> > Thank you for your valuable and insightful comments. We have carefully addressed each of your points and made every effort to incorporate your suggestions into the revised version of our paper. We are confident that our responses have resolved the concerns you raised.
> >
> > We would greatly appreciate it if you could kindly review our responses and consider reassessing our submission. As the discussion phase is concluding soon, we hope to engage in a productive dialogue with you before it ends.
> >
> > Thank you once again for your time and consideration.
> >
> > Thanks
> >
> > Authors

---

> > > ### Author Response · Authors · 2025-08-08
> > > **Requesting Reviewer for Checking Our Responses**
> > >
> > > Dear Reviewer,
> > >
> > > As only 24 hours are left till the discussion period ends, we once again request you to kindly check our responses. We have carefully addressed each of your points and made every effort to incorporate your suggestions into the revised version of our paper. We are confident that our responses have resolved the concerns you raised.
> > >
> > > We would greatly appreciate it if you could kindly review our responses and consider reassessing our submission.

---

> > > > ### Comment · Area_Chair_oJAd · 2025-08-08
> > > > **Last reminder to engage in the discussion**
> > > >
> > > > Dear Reviewer kZ3Z,
> > > >
> > > > The discussion period is now within 24 hours. Please check out the authors' response ASAP!
> > > >
> > > > Best,
> > > > AC

---

### Comment · Area_Chair_oJAd · 2025-08-03
**Reminder: Discussion Phase (July 31 – Aug 6)**

Hi everyone,

This is a reminder that the discussion phase is between July 31 – Aug 6.

Please read the author responses, especially where you are mentioned, and post your reply as soon as possible. This helps ensure there's time for meaningful back-and-forth.

Thanks for your engagement!

AC

---

### Author Response · Authors · 2025-08-05
**Summary of Key Contributions**

We sincerely thank all the reviewers for their thoughtful, constructive, and detailed feedback, which helped us improve the clarity and scope of our work. We use this opportunity to highlight core contributions of our paper -

# Importance of value-guided KV compression

We observe that most existing KV compression methods like SnapKV and H2o use only attention keys for identifying the key-value pairs for eviction during prefill or generation. Over-reliance on only key tensors lead to high post-eviction (reconstruction) loss for these methods. We theoretically justified how attention value-guided methods can ensure lower reconstruction loss, therefore, ensure better preservation of critical key-value pairs. Based on this observation, we design CurDKV, a novel method that compresses KV cache based on leverage scores computed from CUR matrix decomposition.

# Strong empirical results

Experimental results on 16 longbench tasks and 8 needle-in-a-haystack (NIAH) tasks highlight the strong performance of CurDKV with LLaMA and Mistral models. CurDKV consistently outperform the contemporary baselines like SnapKV and ChunkKV with an average margin of 9%.

During the rebuttal phrase, we conducted additional experiments with Qwen-14B and Qwen-32B models, where CurDKV demonstrated superior performance, outperforming the baselines. Even for 128K context length NIAH tasks, CurDKV remains performant under high compression regime, often outperforming the existing KV compression methods.

The ablation studies further illustrate the importance of the design choice of CurDKV.  Our method is also compatible with adaptive budget allocation, further extending its utility for different compression needs.

# Computational benefits of CurDKV

We evaluated CurDKV and other baselines in terms of prefill and generation time on different context and generation length at various compression ratios. Although the prefill time (static) of CurDKV is marginally higher than SnapKV, the generation time is significantly smaller as compared to SnapKV or ChunkKV. The difference is even more striking for longer generation lengths, highlighting the practical usefulness of our method. Moreover, CurDKV is compatible with FlashAttention, Grouped Query Attention and offers seamless integration support with popular libraries like Nvidia's KVPress.

---

### Comment · Area_Chair_oJAd · 2025-08-05
**Please respond to authors' rebuttal**

Dear reviewers,

As the deadline of discussion is approaching, please read the authors' response, check if they address your raised points and provide the feedback accordingly. Many thanks for your cooperation.

Best, AC

---

### Note · Authors · 2025-08-14

We sincerely thank all the reviewers for their thoughtful, constructive, and detailed feedback, which helped us improve the clarity and scope of our work. Here are the highlights from our rebuttal and discussion with each reviewer.

# Key contributions of our papers: as acknowledged by reviewers

## Theoretically strongly motivated KV compression method

Reviewer kZ3Z and hDao acknowledged the importance of value-based KV compression methods and justified the theoretical significance of our proposed method.

## Strong empirical results

Reviewer kZ3Z and NWDK acknowledged the strong empirical performance of CurDKV on 16 longbench and NIAH tasks.

## Strong computational benefits

All reviewers (kZ3Z, 1huD, NWDK, and hDao) applauded CurDKV as compatible with techniques such as FlashAttention and GQA. They acknowledged the practical importance of our proposed method.

## Easy-to-follow paper

Reviewer NWDK also admitted that our paper is easy to follow and well-written.

# Summary of Rebuttal process

* As suggested by reviewer kZ3Z, we conducted additional experiments on the RULER 128K dataset. Even on these longer context tasks, CurDKV outperforms the baselines by more than 1% margin.
* Considering the suggestions by reviewer 1huD, we conducted experiments on four longbench tasks with two larger LMs, Qwen-14B and Qwen-32B. CurDKV exhibits superior performance for both models, beating the other baselines with a significant margin.
* Performed ablation study on Gaussian projection rank ($r$) and safeguard $\alpha$ to demonstrate the importance of these hyperparameters on CurDKV.
* Suggested by reviewers kZ3Z, NWDK, and hDao, we highlighted the prefill time and generation time of CurDKV for different contexts and generation lengths. The additional results highlight the computational efficiency of CurDKV, particularly while generating more extended sequences.

We will integrate these clarifications, ablations, and results into the revision.

# Reviewer response on rebuttal

* Reviewer 1huD and hDao acknowledged that our rebuttals have addressed their concerns.
* Despite multiple reminders, we unfortunately could not receive any acknowledgement from reviewer kZ3Z and NWDK.

We also thank the AC and SAC for their oversight throughout the review process.

Authors

---

### Decision · Program_Chairs · 2025-09-17

**Decision:**

Accept (poster)

**Comment:**

The paper proposes CurDKV, a value-guided KV-cache compression method for LLMs that uses approximated CUR decomposition. Major concerns such as evaluation on longer sequences and latencies are successfully addressed by the authors. All reviewers are positive toward accepting this paper. Hence I vote for acceptance for this paper.